ecology/evolution/biomechanics

cuticle, ridge, *Hevea brasiliensis*, ontogeny, replica, plant–insect interactions

**Author for correspondence:**
Venkata A. Surapaneni
e-mail: amarnadh.sv@bio.uni-freiburg.de

# Spatio-temporal development of cuticular ridges on leaf surfaces of *Hevea brasiliensis* alters insect attachment

Venkata A. Surapaneni[1,2,3], Georg Bold[1,2,3], Thomas Speck[1,2,3,4] and Marc Thielen[1,2,3]

[1]Plant Biomechanics Group, Botanic Garden, Faculty of Biology, University of Freiburg, Schänzlestrasse 1, 79104 Freiburg, Germany
[2]FIT, Freiburg Center for Interactive Materials and Bioinspired Technologies, University of Freiburg, Georges-Köhler-Allee 105, 79110 Freiburg, Germany
[3]FMF, Freiburg Materials Research Center, University of Freiburg, Stefan-Meier-Strasse 21, 79104 Freiburg, Germany
[4]Cluster of Excellence livMatS@ FIT—Freiburg Center for Interactive Materials and Bioinspired Technologies, University of Freiburg, Georges-Köhler-Allee 105, 79110 Freiburg, Germany

VAS, 0000-0002-6241-9048; GB, 0000-0002-8020-8770; TS, 0000-0002-2245-2636; MT, 0000-0002-7773-6724

Cuticular ridges on plant surfaces can control insect adhesion and wetting behaviour and might also offer stability to underlying cells during growth. The growth of the plant cuticle and its underlying cells possibly results in changes in the morphology of cuticular ridges and may also affect their function. We present spatial and temporal patterns in cuticular ridge development on the leaf surfaces of the model plant, *Hevea brasiliensis*. We have identified, by confocal laser scanning microscopy of polymer leaf replicas, an acropetally directed progression of ridges during the ontogeny of *Hevea brasiliensis* leaf surfaces. The use of Colorado potato beetles (*Leptinotarsa decemlineata*) as a model insect species has shown that the changing dimensions of cuticular ridges on plant leaves during ontogeny have a significant impact on insect traction forces and act as an effective indirect defence mechanism. The traction forces of walking insects are significantly lower on mature leaf surfaces compared with young leaf surfaces. The measured walking traction forces exhibit a strong negative correlation with the dimensions of the cuticular ridges.

# 1. Background

The plant cuticle is a continuous extracellular membrane covering the surfaces of most of the above-ground organs of land plants. As an interface between plant epidermal cells and the surrounding environment, it incorporates a highly multifunctional structure necessary for plant survival. The cuticle serves as a transpiration barrier protecting the underlying tissues against water loss and regulates gas exchange together with other functions such as UV light reflection, mechanical integrity, control of water repellence, particle adhesion and animal interactions [1–3]. The mesoscale morphology on the peripheral side of the cuticle is highly diverse among the different plant species and includes hairs, branched or unbranched trichomes, cuticular ridges and valleys (also referred to as wrinkles or striations) and epicuticular waxes [4,5]. These structures in addition to the various underlying cell shapes and to the plant surface chemistry influence abiotic interactions with water and light [6] and biotic interactions with pollinators, herbivores and prey [7]. The cuticle structure is dynamic and thus changes continuously during the ontogeny of a plant, because of new material deposition and the growth of the underlying epidermis [8]. This results in ontogenetic changes in the microscale morphology of plant surfaces and probably also influences their functional properties, for example, water repellence or herbivore interactions.

Whereas the chemical resistance of the plants and digestive ability of herbivores have been the main subjects of interest to assess plant defensive behaviour during ontogeny [9–11], changes in the morphology of cuticular microstructures as a mechanism of structural defence has not as yet been studied in detail [12–14]. For instance, in addition to the chemical resistance and chewing ability of the insects and prey or food availability, insect herbivory or mutualism is also dependent upon the success with which insects are able to hold on to or walk over plant surfaces. An insect's ability to adhere can be reduced on plant surfaces exhibiting epicuticular waxes, cuticular ridges and trichomes as compared with those having smooth surfaces [15–21]. Cuticular ridges are of particular interest from a biomimetic point of view because they are mechanically relatively robust compared with wax-like structures [20,22] and because of the ease of their production [23]. Artificial periodic and randomly oriented ridges have been demonstrated to reduce insect attachment, with the lowest walking forces occurring at ridge dimensions in the critical roughness range of 0.3 to 3 µm [24,25]. Indeed, rough surfaces with various morphologies and surface chemistry [20–22,26–30] show reduced insect adhesion forces in this roughness range. Thus, the knowledge of whether the cuticular structures on plant leaves also display this critical roughness and of the ontogenetic stage at which this is achieved is of interest.

The goal of this study has been to obtain an understanding of (i) the point at which the cuticle ridges appear during the growth of the leaves, (ii) the changes in ridge morphology during growth, and (iii) the way in which these growth-induced changes of plant surfaces affect insect walking behaviour. To study the ontogenetic development of ridges on leaf cuticles, we chose the Pará rubber tree (*Hevea brasiliensis*) as a model plant. Previous literature [21,22] and our personal observations have shown that *H. brasiliensis* has uniformly and densely distributed cuticular ridges on almost flat epidermal cells of the adaxial leaves and is therefore an ideal candidate for the plant model. Originally from the Amazon basin and now grown widely in South and Southeast Asia, this plant is a tropical species with tri-foliated leaves and is a prime source of natural rubber [31]. Rao [32] first reported the presence of cuticular ridges on the adaxial side and reticulate structures on the abaxial side of *H. brasiliensis* leaves. Here, we make use of polymer leaf replicas to study the morphological changes of these cuticular ridges during ontogeny. The polymer replication of leaf surfaces provides a great practical advantage in studying the variations in surface microstructures, as it avoids difficulties encountered because of slowly dehydrating leaf samples [3]. The replicas are useful for carrying out repeatable and long-term studies, especially for young leaf samples that have translucent cuticles, and they can be stored for further experiments. In particular, the confounding variables associated with surface chemistry and chemical changes associated with defensive mechanisms that might occur during the experiments can be eliminated or controlled. For the analysis of insect walking forces, we selected the Colorado potato beetle (*Leptinotarsa decemlineata*), which has hairy attachment systems, as a model species because of its availability and ease of experimentation. It is also a well-studied model for insect traction experiments on rough surfaces [20,21,27] which makes it appropriate for comparison. More importantly, the attachment structures of Colorado potato beetles are similar to those of other relevant insects, for example, Mupli beetle (*Luprops tristis,* a common herbivore of *H. brasiliensis* leaves) [33]. The selection of these plant and insect model species in this study provides a general insight into how the walking abilities (measured as traction forces) of beetles vary with the ontogenetic growth-induced changes in plant cuticular structures. By means of confocal microscopy observations and traction force experiments, we show that the changing dimensions of the cuticular ridges on plant leaves during ontogeny have a significant impact on insect walking behaviour.

# 2. Materials and methods

## 2.1. Leaf collection

Fresh leaves were collected for experiments from the model species, *H. brasiliensis* (tree greater than 10 m height) cultivated in the tropical greenhouse of the Botanic Garden of the University of Freiburg (average temperature = 28°C, average humidity = 57%). The leaves were monitored during growth from bud burst to adult stages, with fresh leaves for the experiments being cut at five defined ontogenetic stages. Preliminary trial experiments were conducted using a confocal laser scanning microscope to find a correlation between leaf growth, changes in colour and morphology of the adaxial leaf surfaces. Following these experiments, the leaf growth stages were determined based on a combination of microscale morphological changes (which also coincided with visual appearances) of the adaxial leaf surfaces and leaf age. The classification of the stages is given in electronic supplementary material, table S1. The leaves at *'stage 1'* (S1, age: 13 ± 2 days after bud burst) were shiny brown in colour and did not have microscale ridges on their surfaces. These leaves were just about to transform from a smooth cellular surface to a ridged surface evident from the appearance of the pale green colour at the very base of the leaves. In order to have sufficient temporal resolution of the growth stages, we defined a transition stage as *'stage 2'* where the transition from a smooth cellular surface to a surface with high aspect ratio ridges occurs. The *'stage 2'* leaves (S2, age: 15 ± 3 days) had half transformed surfaces with ridges covering 40–60% of the leaf surface starting from the base of the leaves. These leaves were again divided into *'stage 2A'* (S2A) and *'stage 2B'* (S2B). S2A refers to the area towards the apex having no ridges and a brown surface; S2B defines the area towards the leaf base with ridges and a pale green surface. *'Stage 3'* leaves (S3, age: 16 ± 3 days) had ridges on the entire area of the leaf surface. The growth of the leaves as estimated from the length of the midrib (electronic supplementary material, figure S1) ceased at *'stage 4'* (S4, age: 21 ± 4 days). *'Stage 5'* (S5) leaves had an age greater than 60 days. Once the leaves were cut, the cut ends were immediately sealed with petroleum jelly (Vaseline) in order to prevent dehydration before the experiments. The replication process (§2.2) of the leaf surfaces was started within 10 min after the leaves were cut. Before replication, the leaves were cleaned gently using deionized water and carefully dried with compressed air.

## 2.2. Surface replication

For leaf surface replication, an Epoxy-PDMS replication method as described in Kumar *et al.* [34] was used, except that entire leaf laminas (minimum leaf lamina area = 1168 mm$^2$, maximum leaf lamina area = 14 383 mm$^2$) were replicated in our study. The leaves were first attached to a clean flat plate by using double-sided adhesive tape (Tesa SE, Norderstedt, Germany) with the adaxial side of the leaves facing upwards. The perimeter of the leaf was framed with poly-vinyl siloxane (PVS, President Light Body, ColteneWhaledent, Altstätten, Switzerland), a common imprinting material for dentistry with fast polymerization times of around 10 min, in order to create a casting mould.

For negative replicas, a two-component epoxy resin (Epoxy Resin L & Hardener S, Toolcraft, Conrad Electronic SE, Hirschau, Germany; resin to hardener mixing ratio of 10 : 4.8) was uniformly mixed in a plastic cup for about 5 min using a glass rod. After being degassed in a vacuum chamber for 15–20 min to remove trapped air bubbles, the epoxy resin was carefully poured onto the leaf surface. Following overnight curing (approx. 15 h) of the resin at room temperature, the leaves were carefully peeled off these now-cured epoxy negative moulds. If necessary, potassium hydroxide solution (KOH, ≥85%, p.a., Carl Roth GmbH & Co. KG, Karlsruhe, Germany; concentration: 60 g/100 ml) was used to separate the leaves from the epoxy moulds, the reasons for which are discussed later. In this case, the epoxy mould with leaf remnants sticking to it was placed in a KOH solution at 60 ± 3°C with a magnetic stirrer running at 450 ± 25 r.p.m. for 20 h, followed by ultrasonication (in deionized water) for 10–15 min.

For positive replicas, a two-component polydimethylsiloxane (PDMS) elastomer (Bluesil ESA 7250 A & B kit, Bluestar Silicones GmbH, Leverkusen, Germany; weight ratio of 10 : 1) was used. A drop of red dye (Holcosil LSR Red, Holland Colours Europe BV, Apeldoorn, The Netherlands) was added to reduce transparency and thus to ease microscopic observations. The components were uniformly mixed in a plastic cup for about 5 min and degassed until the air bubbles in the mixture were removed. The mixture was then slowly poured onto the negative moulds and kept in a vacuum chamber for 1 h to remove any remaining air entrapped around the sample. The air-free mixture was then placed in an oven at 60°C for 4 h for curing. The cured replicas were then gently peeled off and

cleaned in isopropyl alcohol (≥99.95%, Carl Roth GmbH & Co. KG, Karlsruhe, Germany) in an ultrasonicator for 10 min, followed by drying. In total, five leaves within each ontogenetic leaf stage were replicated. For comparisons, five clean glass slides were also replicated, and henceforth, we term the replicas of glass as 'PDMS glass replicas'. The efficiency of replication of young and adult *H. brasiliensis* leaves was as quantified in Kumar *et al.* [34] to which we also refer the interested reader for further details concerning the process.

## 2.3. Surface characterization

The replicas of fresh leaf samples were characterized using a confocal laser scanning microscope (CLSM, Olympus LEXT OLS4000, 405 nm laser, Olympus Corporation, Tokyo, Japan). On each leaf replica, a total of 10 randomly selected spots of $65 \times 65$ µm on both sides of the midrib were recorded at 4320-fold (50x objective) magnification. A commercial surface analysis software (Mountains Map Premium version 7.4, Digital Surf SARL, Besançon, France) was used to analyse the measurements from CLSM. Following the application of a median noise filter, zig-zag profile lines of at least 200 µm in length were taken on the topographic layer of each spot. A standard Gaussian filter was applied to the profiles in order to separate waviness and roughness (2.5 µm for glass, PDMS glass replicas, S1, S2, S3; 8 µm for S4 and S5). Line roughness parameters *Rc* (mean height), *Ra* (arithmetic average of the roughness), *Rsm* (mean spacing) and *Rsk* (Skewness) were then calculated from the roughness profiles. Each cuticular ridge on the leaf replica is considered as a profile element having one peak and one valley. When measured over the sampling length (i.e. 200 µm), the mean height of the profile elements (*Rc*) represents the mean value of the heights of profile elements within the sampling length. Similarly, the mean spacing of the profile elements (*Rsm*) represents the mean value of the length of the profile elements within the sampling length. The arithmetic average of the roughness (*Ra*) is one of the most widely used parameters for roughness measurement. It is the arithmetic average of the absolute height deviations of the roughness from the mean line. The skewness of the profile (*Rsk*) measures the symmetry of the surface deviations about the mean line. It is the ratio of mean cube value of height of the profile elements and the cube of the root mean square deviation. A value of *Rsk* = 0 represents symmetric profile elements about the mean line. A value of *Rsk* > 0 means the presence of sharper peaks and larger valleys and a value of *Rsk* < 0 means the presence of blunt peaks with sharper valleys. For a detailed scientific background of the roughness parameters, the interested reader is referred to any standard textbook on engineering metrology [35]. For ease of comparison, the mean aspect ratio (*AR* = *Rc*/*Rsm*) was defined to characterize and compare the changes in ridge dimensions and the effect on insect adhesion. The Fiji image processing package, running IMAGEJ version 1.51u [36], was used for leaf area measurements.

Scanning electron microscopy (SEM) observations of replica samples and beetle tarsal attachment systems were carried out using a Leo 435 VP, Leica, Wiesbaden, Germany, at an accelerating voltage of 15 kV. Prior to SEM visualization, dead beetles were dehydrated in a graded methanol series (50, 70 and 100% methanol each for 24 h) before being critical-point dried (LPD 030, Bal-Tec) in ethanol [37]. The dried specimen together with the leaf replicas were immediately transferred to a gold sputter coater (Cressington Sputter Coater, 108 auto) for 15–20 nm gold-coating before examination in SEM.

## 2.4. Insect species and traction experiments

Female Colorado potato beetles (*Leptinotarsa decemlineata*, Coleoptera: Chrysomelidae) were used as model insect species (with hairy tarsal attachment system) for traction experiments. All the beetles were collected from a potato field (organically farmed) in Kirchzarten area near Freiburg, Germany and kept in a terrarium. The beetles were fed with potato leaves and the lighting conditions were fixed at a day–night regime of 16L : 8D by using a lamp (Osram Lumilux Daylight 860, 58 W). The average mass of the beetles was $182.94 \pm 27.71$ mg. Traction experiments performed in order to analyse the differences in insect adhesion with leaf ontogenetic stage were carried out as described in Prüm *et al.* [22,23]. Maximum walking frictional forces were measured using a highly sensitive force transducer (FORT 25, force range: 0–0.25 N, World Precision Instruments Inc., Sarasota, USA) that was attached to the elytra of each beetle via human hair by using a tiny drop of molten beeswax. Forces were recorded for at least 2 min of active walking at an ambient mean temperature of $25.0 \pm 0.8$°C and humidity of $55.0 \pm 5.2$% RH. Only data from walking in a straight line with less than a $\pm 2$° variation were taken into consideration. Within each measurement, i.e. the force over time curve, the 15 highest local maxima with a minimum interval of 3 s between neighbouring peaks were identified and the

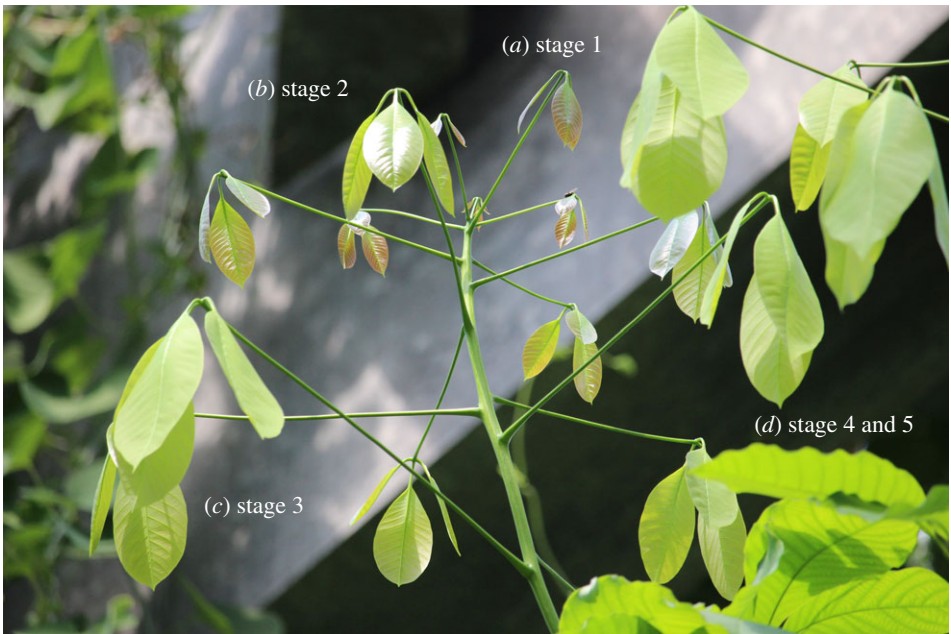

**Figure 1.** Leaf transition: leaves from the tip of a branch of *Hevea brasiliensis* tree (height greater than 10 m) with almost vertically drooping leaves during young stages; the leaves gradually position themselves more horizontally as they grow. The image shows (*a*) young leaves at stage S1, (*b*) leaves in transition to stage S2 with the surface progressing from shiny brown to dull pale green acropetally, (*c*) leaves at stage S3 and (*d*) adult leaves (S4 and S5).

median of these 15 maxima was extracted. The measurements were repeated with eight different beetles on each replica. Electronic supplementary material, figure S2 shows a schematic of the experimental set-up used for the traction experiments; the supplementary video shows a beetle walking on the replica surface.

## 2.5. Statistics

Statistical comparisons were performed by an ANOVA on a generalized linear mixed model (GLMM) using the *R* software environment version 3.6.1 [38] and the *lme4* package [39]. Both the original and ranked data had a non-normal distribution and non-homogeneity in variances. The experimental design included repeated measurements and five replicates (leaf samples) for each leaf stage. Thus, GLMM with a gamma distribution and log link was used to test the differences in traction forces with growth stages as a fixed factor and with replicates (leaf samples) and beetles as random factors. Before the comparisons, one outlier adjustment on a single observation of glass (out of 40, glass) was made with the rounded value of the median. For comparisons of *Rc*, *Ra*, *Rsm* and *AR*, GLMM with a gamma distribution (log link) was used with replicates as random factors and the leaf area as the covariate. *Post-hoc* tests were conducted using multiple comparison of means with Tukey contrasts by using the *emmeans* package [40]. Values of $p < 0.05$ were considered to be statistically significant.

# 3. Results

## 3.1. Ontogeny and cuticular ridge development

The leaves of the studied *H. brasiliensis* tree (figure 1) reached growth stage S1 in 11 to 15 days from the formation of buds. The acropetally directed transition of the leaf surfaces in the form of striking colour changes occurred immediately after S1 during the transition to S2 in 1 to 3 days. Once the transition from S2 to S3 (1 day) was complete, the leaves continued to grow exponentially (electronic supplementary material, figure S1) until reaching S4, which lasted for 4 to 6 days. Thereafter, no significant growth of the leaves was recorded. The changes in the colour of the leaves with growth stages and the CLSM observations at each stage are shown in figure 2. After *ca* 60 days from bud formation, the leaves (S5)

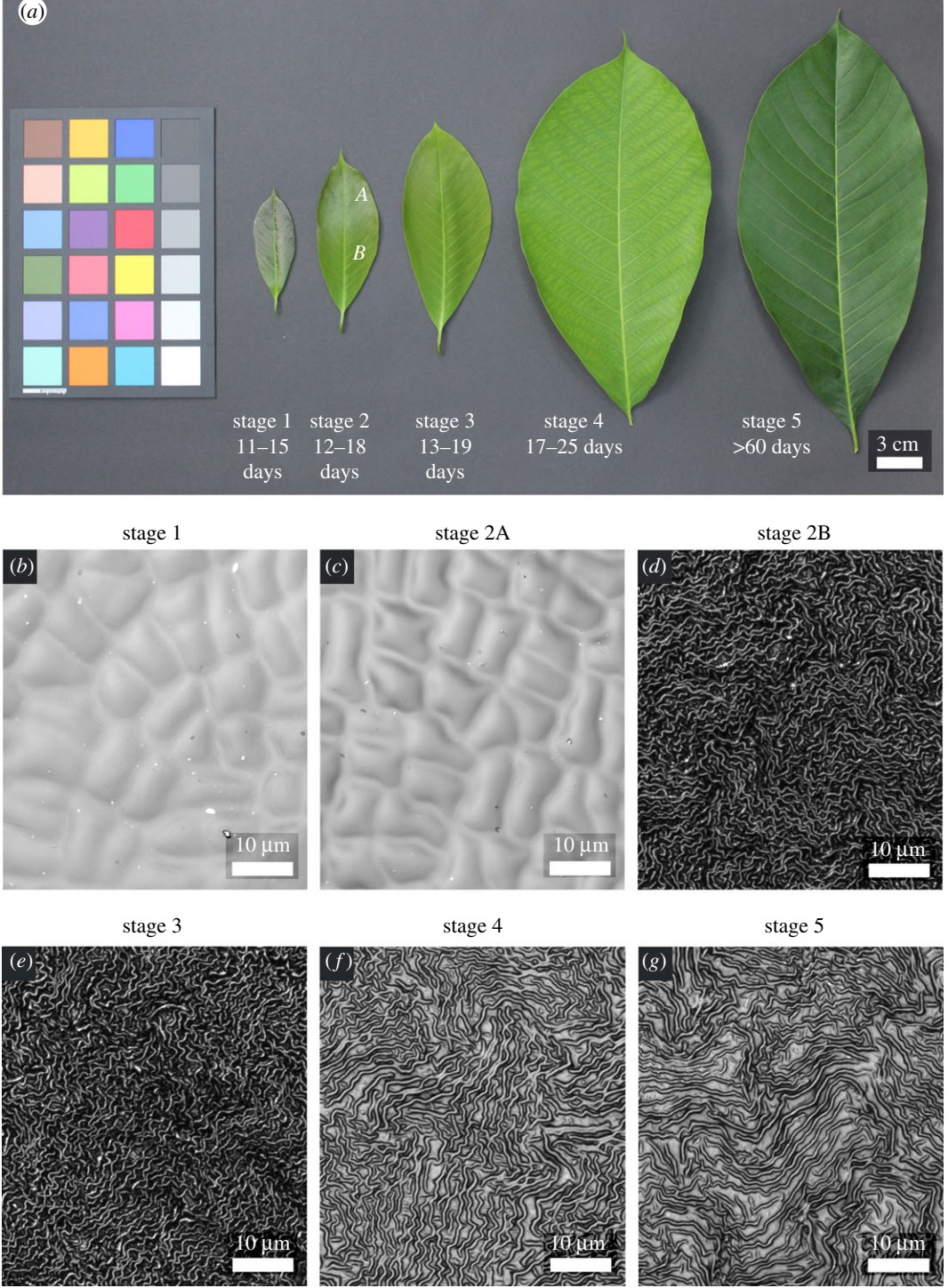

**Figure 2.** Development of leaves and of cuticular ridges: (*a*) *Hevea brasiliensis* leaves at various growth stages with a colour spectrum (SpyderCHECKR 24, SCK200, Datacolor AG Europe, Rotkreuz, Switzerland). (*b*–*g*) Corresponding CLSM images of the leaf replicas at various stages: (*b*) stage 1, (*c*) stage 2A, (*d*) stage 2B, (*e*) stage 3, (*f*) stage 4 and (*g*) stage 5. The appearance of the pale green colour towards the basal region of leaf at stage 2B coincides with the appearance of microscale ridges. Both the colour and the ridges progress acropetally and further develop during the following growth stages.

exhibited apparently reduced hydrophobic behaviour and dust particles accumulated on their surfaces; this was not observed on leaves at earlier growth stages. Throughout the stages, the colour variation of the leaves was visually evident with the dark brown colour (S1 and S2A) during the initial stages of leaves gradually changing to pale green with a colour gradient moving from the base of the leaves

**Table 1.** Median values of the roughness parameters from all the replicates of glass, PDMS glass replicas and leaf replica surfaces. For stages S2B and S3, values from clean small areas are given representing the realistic high aspect ratio of ridges.

| surface | age (days) | $Ra$ (µm) | $Rc$ (µm) | $Rsm$ (µm) | aspect ratio $= Rc/Rsm$ | $Rsk$ |
|---|---|---|---|---|---|---|
| glass | — | 0.0004 | 0.001 | 1.020 | 0.001 | 0.346 |
| PDMS glass replica | — | 0.0005 | 0.002 | 1.250 | 0.002 | 0.328 |
| S1 | 13 ± 2 | 0.008 | 0.031 | 4.465 | 0.006 | −0.906 |
| S2A | 15 ± 3 | 0.010 | 0.038 | 4.440 | 0.008 | −0.768 |
| S2B | 15 ± 3 | 0.111 | 0.358 | 1.095 | 0.323 | −0.265 |
| S3 | 16 ± 3 | 0.132 | 0.418 | 1.13 | 0.378 | −0.27 |
| S4 | 21 ± 4 | 0.181 | 0.542 | 2.105 | 0.261 | 0.215 |
| S5 | >60 | 0.190 | 0.579 | 2.020 | 0.283 | 0.035 |

towards the apex (S2B and S3) before finally shifting to dark green (S5) (figure 2*a*). It is conspicuous how the colour variations of the leaves coincided with the formation and development of the cuticular ridges on the leaf surfaces.

The CLSM measurements of the replica surfaces (figure 2*b–g*, electronic supplementary material figure S3) and the results from initial experiments revealed that, until stage S1, the leaf surfaces of *H. brasiliensis* were composed of smooth cells free of any superimposed structuring. Thereafter, a rapid development of ridges during S2 could be observed, progressing acropetally from the base to the apex of the leaves. At this stage, the area towards the apex (S2A) still contained smooth cells similar to those of S1. At the stages S2B and S3, the cells were densely covered with cuticular ridges having a high aspect ratio of height to spacing (table 1). This resulted in the firm sticking of the original leaf material to all epoxy negative replicas (moulds) of these stages and KOH treatment was required in order to remove the plant material from the surfaces. Nevertheless, thin residues of plant material remained fused to the epoxy moulds over large leaf areas (figure 3), even after KOH treatment and ultrasonication, thereby allowing only relatively small spots of the leaf surfaces to be properly replicated. Electronic supplementary material, figure S4 shows an SEM image of an epoxy replica with plant material still sticking to parts of it. This contamination also influenced the subsequent fabrication of positive replicas, resulting in a significant reduction of the real aspect ratio of the ridges (electronic supplementary material, figure S5 (a,b) and electronic supplementary material, table S2). However, for the morphometric comparison, we were still able to use the data from tiny spots (1 to 14 spots with areas around 2500 to 6400 µm$^2$) in which the epoxy moulds were not tainted with plant residues. On these spots, the real aspect ratios were measured. The roughness parameters of these spots are included in table 1 and figure 4. The topology parameters of positive replicas from the larger tainted negatives of stage S2B and S3 are also given in electronic supplementary material table S2 and figure 4 (boxplots in grey). As the untainted spots of the positive replicas were too small for mechanical testing, we could only perform traction force measurements of stages S2B and S3 on positive replicas from the large tainted negative moulds, despite being well aware that these data, because of the markedly reduced aspect ratio of the ridges, probably overestimated the real traction forces on surfaces of these stages (figure 5). Following stage S3, the ridges developed to give a labyrinth-type arrangement over the entire area of the leaves with reduced height and increased spacing until S4 during which the leaf growth ceased. At stage S5, the overall morphology of the ridges remained similar to that of S4. These observations revealed three distinct levels of microstructural surface morphology on the adaxial side of the leaves of *H. brasiliensis*: (i) surfaces with a smooth cellular structure exhibiting no ridges from the bud appearance until stages S1 and S2A, (ii) high aspect ratio ridges at stages S2B and S3, and (iii) the labyrinth-type arrangement of developed ridges at stages S4 and S5. An exception to this was the very basal part of the leaves at stages S2B and S3. Here, the ridges were fully developed and their morphology was similar to that of adult stages 4 and 5. Accordingly, the removal of the plant material from the epoxy negative was much easier in these regions (figure 3). Electronic supplementary material figure S5 (c) shows such a region on one of the replicas at S3.

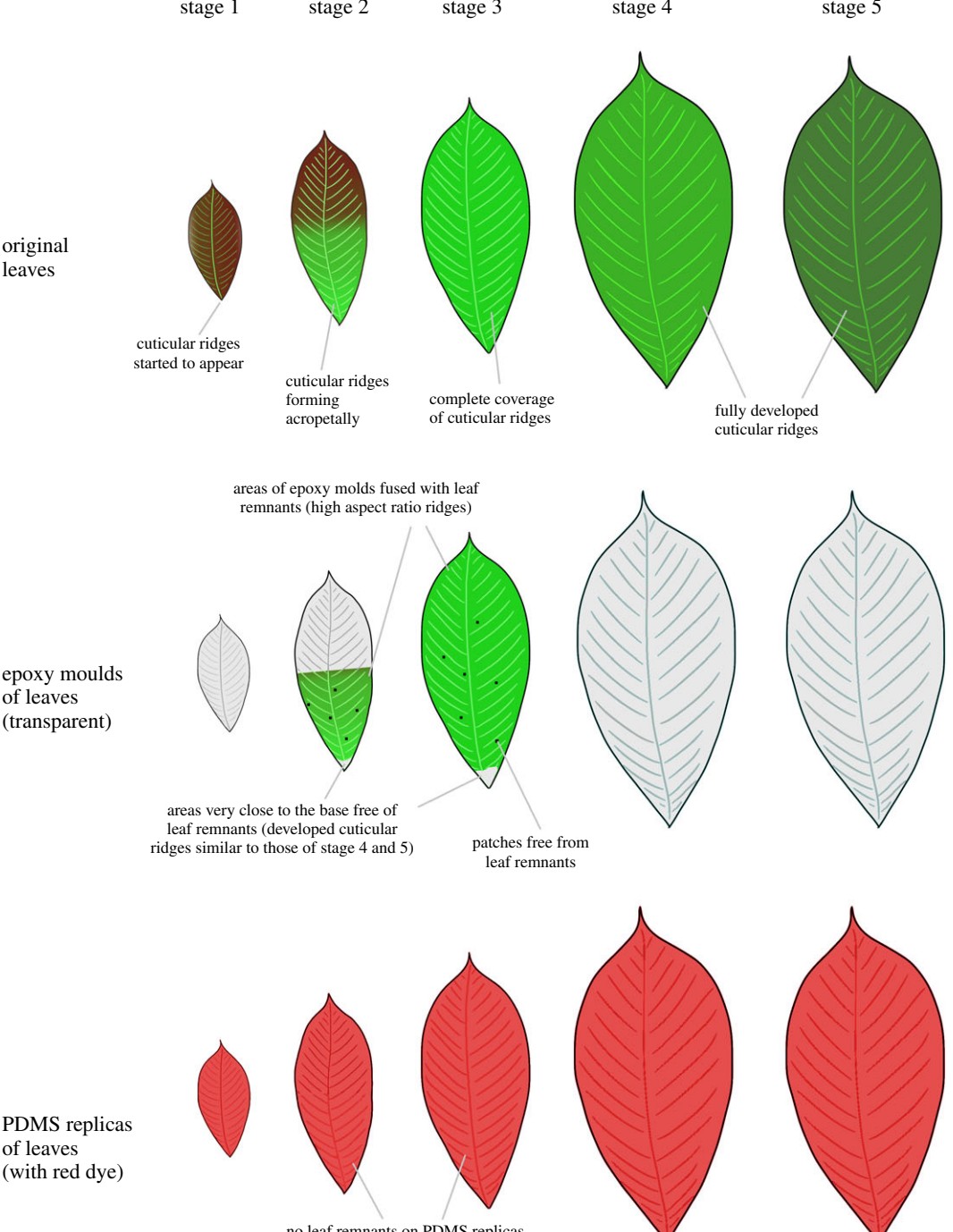

**Figure 3.** Leaf replication: schematic representation of original leaves and their negative (epoxy) and positive (PDMS) replicas. A large portion of the epoxy replicas of stages S2B (basal region of the stage 2 leaves) and S3 was fused with leaf remnants. A few patches and the region close to the base of the leaves at stages S2B and S3 were free of leaf remnants. The surfaces of the PDMS positive replicas did not have any leaf remnants.

## 3.2. Ridge dimensions

In order to quantify the differences in surface structuring between the leaf stages, we compared the roughness parameters $Rc$, $Ra$, $Rsk$ and $Rsm$ assessed from leaf replicas. The variations in roughness are shown as ridge aspect ratio ($AR = Rc/Rsm$) versus leaf stage in figure 4. The roughness values of the replicas of leaves from all the stages compared with glass and its replica (i.e. PDMS glass replica) showed significant differences (ANOVA type III on GLMM, $p < 0.001$, $n = 369$). The median values of these roughness parameters on five replicates of glass, of PDMS glass replica and of leaf replica surfaces are given in table 1.

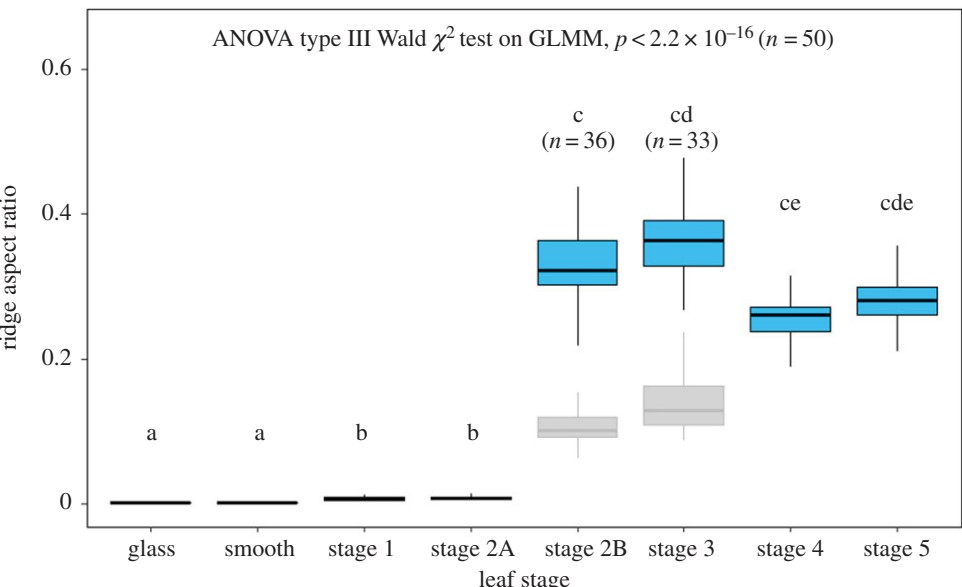

**Figure 4.** Ridge aspect ratio versus leaf stage: boxplot showing variation in the ridge (roughness) aspect ratio for glass, PDMS glass replicas and replicas of leaves at various leaf stages. For stages S2B and S3, realistic aspect ratio values are given for clean replicas from tiny patches and for the replicas from contaminated moulds representing the remaining surface, which were used for traction experiments (box plots in grey; they underestimate the aspect ratio) (see Results section).

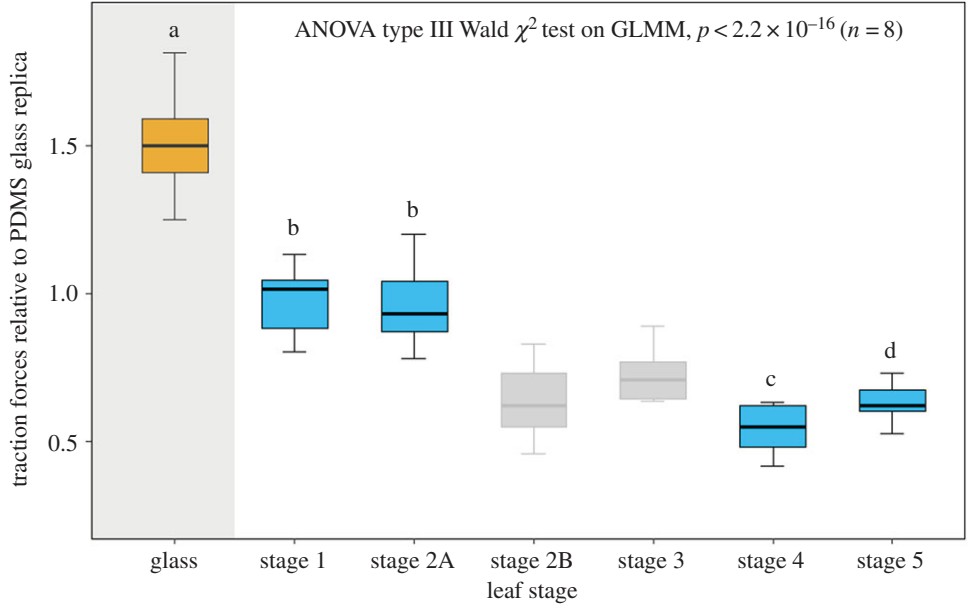

**Figure 5.** Insect traction forces: boxplot showing traction forces of *Leptinotarsa decemlineata* (*n* = 8) on glass and PDMS replica surfaces of leaves at various growth stages relative to traction forces on PDMS glass replicas. The traction force values in the boxplots were normalized to the forces measured on PDMS glass replicas in order to allow comparison with leaf replicas made from the same PDMS material. The traction force values for stage 2B and stage 3 could only be calculated from replicas from contaminated moulds (box plots in grey) and therefore overestimate the real values (see Results section).

The variations in roughness were compared using the arithmetic average of the roughness (*Ra*) and mean aspect ratio of the ridges (*AR* = *Rc*/*Rsm*). The aspect ratio and *Ra* of the ridges did not vary between glass and PDMS glass replicas (*AR*: *p* = 0.971, *Ra*: *p* = 0.446, *n* = 50) but varied significantly between PDMS glass replicas and S1 leaf replicas (*p* < 0.001, *n* = 50). The AR and *Ra* values did not change significantly from S1 to S2A leaf replica surfaces (*AR*: *p* = 0.894, *Ra*: *p* = 0.644, *n* = 50). The roughness parameters of S2B were significantly higher (*AR*: *p* < 0.001, *Ra*: *p* < 0.001, *n*, S2A = 50, *n*, S2B = 36) compared with S2A (note that only untainted spots were used for these measurements). No

significant differences were detected in the roughness parameters between S2B and S3 (AR: $p = 0.619$, $Ra$: $p = 0.314$, $n$, S2B = 36, $n$, S3 = 33). Stages S3 to S4 exhibited no significant differences in the Ra values ($p = 0.903$, $n$, S3 = 33, $n$, S4 = 50) but the aspect ratio of the ridges decreased significantly ($p = 0.007$, $n$, S3 = 33, $n$, S4 = 50). In spite of the large age difference, no significant differences were observed in the $AR$ and $Ra$ values between S4 and S5 (AR: $p = 0.907$, $Ra$: $p = 0.968$, $n = 50$), showing that growth ceased during stage S4. However, a relatively large difference in the $Rsk$ value was found between these stages (table 1). A higher value of skewness generally indicates the presence of sharper ridges with a larger valley size, whereas a lower value indicates the presence of blunt ridges with a smaller valley size. The skewness parameter changed from 0.215 for S4 leaf replicas to 0.035 for S5 leaf replicas (table 1) with an almost constant spacing between the ridges, thus suggesting relatively blunt ridges on S5 leaves. The negative skewness values of the ridges for stages S2B and S3 indicated a dense arrangement of ridges with fewer valleys. The variation of $Ra$ and $AR$ within replicates (leaf samples) at each stage was less than 15% compared with the residual variance. At least three out of five replicates from each stage showed no significant spatial variation in $Ra$ and $AR$ of the ridges in the direction perpendicular to the midrib and at least four out of five replicates exhibited no significant variation in $Ra$ and $AR$ of the ridges in the direction parallel to the midrib. No significant effect of leaf area on the roughness parameters was observed.

## 3.3. Insect traction forces

Our experiments showed a significant effect not only of ontogenetic changes in leaf surface structure, but also of concomitant insect traction forces. The maximum insect traction forces on all PDMS leaf replicas differed significantly from those on glass (ANOVA type III on GLMM, $p < 0.001$, $n = 240$). We excluded the results of insect traction forces on the S2B and S3 leaf replicas from the statistical analysis for the above-mentioned reasons. An analysis of random effects showed the relatively lower influence of replicates, i.e. the variation within leaves in each stage (4.2%, variance: 0.003, residual variance: 0.038), compared with that from the beetles (27.4%, variance: 0.014, residual variance: 0.038), and the differences in traction forces within replicates in each stage were not significant. Therefore, the means of the traction forces within replicates in each stage were used for further comparisons. Figure 5 shows the variation in the mean traction forces of insects on glass and leaf replicas relative to the traction forces on PDMS glass replicas. Electronic supplementary material, figure S6 shows the variation in absolute traction forces on glass, PDMS glass replicas and leaf replicas with the data on each replicate included ($n = 40$). The traction forces on glass surfaces ($F_{median} = 9.71$ mN) were much higher than on PDMS glass replica surfaces ($F_{median} = 6.75$ mN, $p < 0.001$, $n = 8$) showing the effect of surface chemistry on insect adhesion. Compared with the PDMS glass replicas, the replicas of leaves at S1 ($F_{median} = 6.31$ mN, $p = 0.988$, $n = 8$) and S2A ($F_{median} = 6.91$ mN, $p = 0.934$, $n = 8$) did not differ significantly in insect traction forces. Even though the dimensions of the ridges were influenced by the plant remnants at S2B (replicas from tainted moulds), the traction forces showed a decrease in magnitude compared with those on the surfaces towards the apex (S2A). In reality, the insect traction forces on the leaves at stages S2B ($F_{median} = 4.42$ mN) and S3 ($F_{median} = 4.8$ mN) might even be lower than or comparable with the traction forces on leaves at stage S4, because of the critical roughness (0.3–3 µm) of the cuticular ridges at S2B and S3 stages. The traction forces on the adult leaf replicas (S4 and S5) were significantly lower ($p < 0.001$) compared with those on young leaves (S1 and S2A). The mean traction forces at stage S5 increased significantly ($F_{median} = 4.47$ mN, $p = 0.015$, $n = 8$) compared with the forces at stage S4 ($F_{median} = 3.75$ mN). During the experiments, the interaction of beetles with the protruding veins was unavoidable. Nonetheless, these structures did not seem to affect the reduction of traction forces during leaf growth. The log-transformed mean of the traction force and mean aspect ratio ($AR$) values taken over replicates showed a strong association (Pearson's product-moment correlation, $R = -0.91$, $df = 28$, $p < 0.001$) (electronic supplementary material, figure S7). The association between the log-transformed values of the traction forces was also strong compared with $Rc$ and $Ra$ (both, $R = -0.90$, $df = 28$, $p < 0.001$). However, no correlation was found between log-transformed values of traction forces and $Rsm$ ($R = 0.28$, $df = 28$, $p = 0.18$).

## 4. Discussion

In this study, we explored the ontogenetic changes occurring in the morphology of cuticular ridges on the adaxial leaf surfaces of *H. brasiliensis*. Initial screening using confocal laser scanning microscopy helped to

verify the segregation of the leaf developmental stages S1, S2 and S3 based on the colour differences as they coincided with the drastic changes occurring in the surface microstructure. This differed from the definition of leaf growth stages for the same species in earlier studies, which only involved macroscale morphological parameters or physiological data [41–45]. Based on morphology, for example, Dijkman [41] discerned four leaf growth stages A, B, C and D corresponding to bud burst, young, fully expanded and mature states, respectively. As our work was aimed at understanding the effect of changes in the morphology of surface microstructures on insect walking forces, we based our classification on these microstructures, which, nevertheless, were also reflected to a large extent by macroscopically visible features (i.e. colour). The growth stages S1, S4 and S5 defined in our study corresponded to the stages B, C and D [41], respectively. In addition to these, our definition included intermediate stages S2A, S2B and S3 between B and C, where a sudden transition from smooth to dense ridge morphology occurred on the leaf surfaces simultaneously with the exponential growth of the leaves. Chemical changes were earlier attributed as being involved in the colour variations in *H. brasiliensis* leaves, the shiny brown colour during the initial stages of *H. brasiliensis* leaves being the result of the presence of a higher amount of anthocyanin pigments compared with the adult stages of the leaves [46]. The changes in the thickness of the leaf cuticle together with the cell wall material might also contribute to the variation in colour. Whether the colour-affecting chemical changes also progress simultaneously with cuticle development remains uncertain.

Rapid changes in the cuticular morphology on the adaxial leaf surfaces of *H. brasiliensis* occur directionally with ridges progressing acropetally from base to apex. Such ontogenetic development of ridges has also been observed on *Arabidopsis thaliana* sepals [47] on which the progression of the ridges takes place basipetally from the tip to base and is correlated with the reduced growth rate and termination of the cell division of the underlying epidermal cells [47]. This also seems a probable cause in case of *H. brasiliensis* leaves. In addition to the acropetally progressing formation of ridges during the ontogeny of *H. brasiliensis* leaves, a second process is superimposed: at the stages S2B and S3, stretching and thickening of the originally high aspect ratio ridges starts from the leaf base eventually leading to the typical ridge pattern found in the later stages (figure 2 and figure 3). In these stages, we have not observed significant spatial variation of ridge morphology over the leaf area, except in these basal regions. This suggests that both the ridge formation and the development processes on *H. brasiliensis* leaves are not gradual but occur sequentially, possibly a result of polarity in cell maturation and growth. Further work is needed to confirm this hypothesis. Martens [48] proposed that the formation of ridges is attributable to the over-secretion of cuticle above the growing epidermal cells. Recently, a mathematical model has been proposed indicating that the formation of ridges is a result of the mechanical buckling of the cuticle because of compressive stresses arising from a mismatch in epidermal cell expansion and cuticle production [49]. Once formed, the morphology of the ridges is expected to change with growth. This is of interest as it hints at the timing when cuticle production outpaces the underlying epidermal cell growth or vice versa. Intuitively, once the rate of cuticle production reduces, the cuticle is stretched because of the growth of the underlying epidermal cells. Our measurements on the surfaces of *H. brasiliensis* leaves demonstrate that the ridge development stops at stage S4 when the leaves attain maximum growth and that, thereafter, no significant changes in the ridge aspect ratio occur until stage S5. This indicates that ridge development occurs simultaneously with growth in *H. brasiliensis* leaves, with the sudden formation of ridges in the transition stage (S2) and further development until leaf growth ceases. Changes in the amount and chemistry of cuticular waxes also occur during the growth of leaves and may lead to changes in the hydrophobic behaviour of the leaf surfaces [50–52]. When the hydrophobicity of the leaf surfaces is reduced with age, damage of superficial waxes [53] can also occur, caused by rainwater, dust, etc. under field conditions, thus changing the morphology of the cuticular surface. Our model species was grown in a tropical greenhouse. We observed that the leaves at S5 stage have apparently reduced hydrophobic behaviour when compared with S1 to S4 leaf stages. This reduced hydrophobicity combined with the field conditions in the tropical greenhouse might have resulted in the damage of outer layers of cuticular waxes. This could be a reason that the leaf replica surfaces at stage S5 exhibit a reduced skewness parameter compared with those at stage S4.

Whereas *L. decemlineata* is a pest specific to potato plants, their attachment systems [33] are comparable with those of other similar harmful pests. A comparison of the interaction abilities of insects on *H. brasiliensis* leaves, e.g. leaf eating *Adoretus compressus* (Coleoptera) [54] or litter beetle *Luprops tristis* (Coleoptera) [55,56], might help in the understanding of plant–insect interactions, especially with respect to ontogeny. The litter beetle *L. tristis* has been found to have a minimum preference for mature leaves of rubber trees and feed largely on premature leaf litter [55]. This is because

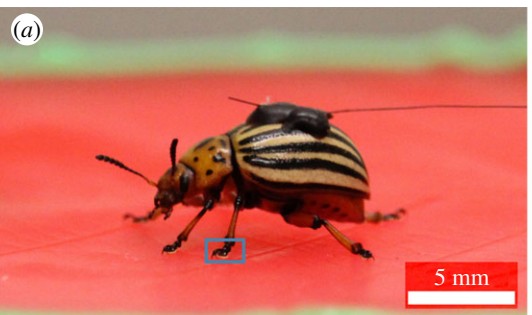

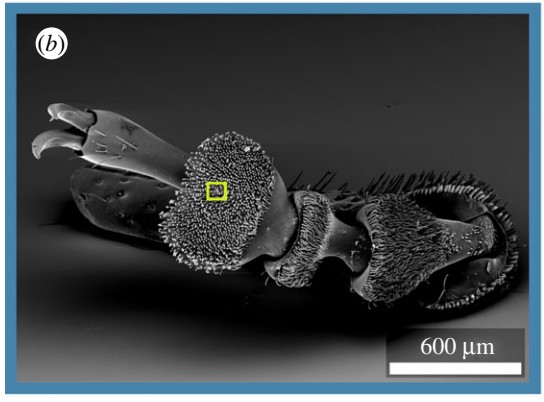

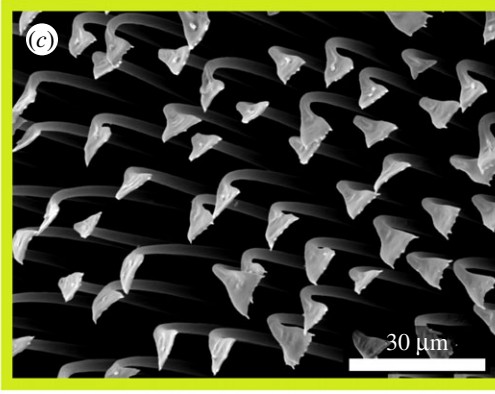

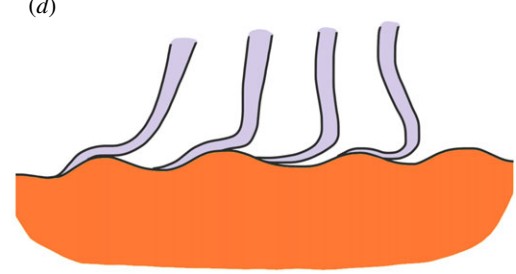

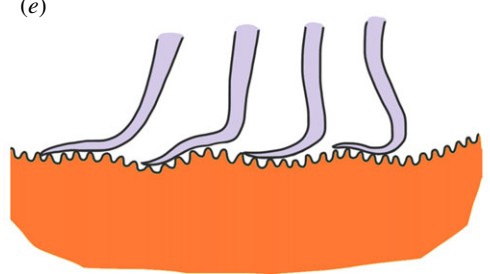

**Figure 6.** Insect adhesion in interaction with plant surface: (*a*) Colorado potato beetle (*Leptinotarsa decemlineata*) walking on a leaf replica, (*b*) SEM image showing tarsal features on the mid leg of female Colorado potato beetle *Leptinotarsa decemlineata*, (*c*) SEM image showing spatula-type terminal ends of the hairy attachment system of the insect. (*d–e*) Schematic representation of the interfacial contact area formation between the terminal ends of insect setae and the leaf surfaces at early (*d*) and adult (*e*) growth stages of *Hevea brasiliensis* leaves.

the leaves of *H. brasiliensis* employ cyanogenic chemical defences (release of hydrogen cyanide, HCN in response to tissue damage) during early growth stages and lignification in adult stages, which are particularly efficient against herbivore and pathogen attack, respectively [45,46,57]. However, insect herbivory on young developing leaves seems to be a global phenomenon. During early stages, plants contain defensive secondary chemicals but also high nutrition values, which decrease in concentration with growth, whereas in the later stages, physical traits such as leaf toughness, non-glandular trichomes and indirect defences, including extra floral nectaries that attract predators, which eat pest insects, have been found to dominate [10,58]. Physical changes occurring in the cuticular microstructures during growth as an indicator of indirect defence have, however, been neglected. Assuming resource allocation and the root-to-shoot ratio as defining criteria for whole-plant defence, Boege & Marquis [9] have proposed that plant defence is lower during the young seedling stage, gradually increases with growth and finally decreases during the over-mature stages. In agreement with this theory, our results also show this pattern in *H. brasiliensis* leaves, with high insect traction forces during young leaf stages, decreasing forces during growth until full expansion and then significantly increasing forces at much later (over-mature) growth stages. The fact that the variables of surface chemistry have been eliminated in our study implies that the dimensional changes in microscale plant surface structures alone can influence the way that insects forage and feed on plant organs. We suggest that similar changes in physical defence traits also occur on other plant species having cuticular structuring. In addition to chemical defences and

internal structural changes such as cell wall hardening and lignification that stiffen plant structure, we argue that changes in cuticular patterning provide a dominating indirect structural response strategy of plants to insect herbivores.

The comparably smooth surfaces of the epidermal cells of the leaves at early ontogenetic stages provide maximum contact area (figure 6d) for the hairy attachment systems of beetles (figure 6a–c) and therefore higher traction forces. On the microscale wrinkled surfaces, however, the possible contact area is reduced (figure 6e) leading to reduced traction forces [20–22]. The traction forces obtained in this study agree well with previous traction force data of Colorado potato beetle on rough plant and artificial surfaces [20,21,27]. Earlier studies have found that a critical roughness of surfaces exists with an asperity size close to 0.3 to 3 µm at which insects fail to maintain proper attachment [26–30,33]. Our data establish that the dimensions of the cuticular ridges on the adult leaves of H. brasiliensis also fall within this critical roughness regime, thus explaining the reduction of insect adhesion forces. This knowledge is important from a biomimetic perspective, since the small changes in the structural parameters of plant surfaces reveal how insects behave on these surfaces. Such knowledge also has great potential in agriculture, as new genetic and technical tools become available for solutions to pest or pollinator management. For example, yet long way to go, studies in genetic modifications [59] in crops in order to induce insect-repelling cuticular structures could help in managing pests in an eco-friendly manner and provide new directions in pest control research. Artificial insect repellent surfaces and coatings inspired by plant cuticular structures may also find applications in agriculture and in daily life [25].

In summary, CLSM measurements on PDMS replicas of H. brasiliensis leaves have demonstrated how microstructural morphology of the leaf cuticle changes spatially and temporally during ontogeny. We have found that the development of the cuticular ridges on H. brasiliensis leaves occurs acropetally and sequentially. Insect adhesion was found to be greatly influenced by the growth-induced morphological changes in leaf cuticular morphology signifying that plant indirect structural defences vary during ontogeny. Changes in insect walking forces have been compared with the dimensional parameters of cuticular structures and the forces were found to be strongly correlated with roughness parameters. Our results provide valuable insights for both technical and genetic agricultural solutions for pest management and are also beneficial in the field of biomimetics for the development of chemical-free insect-resistant surfaces.

Data accessibility. The datasets supporting the results presented in this article are uploaded and available online as .zip files at: https://freidok.uni-freiburg.de/data/166607 (doi:10.6094/UNIFR/166607) [60].
Authors' contributions. M.T. and T.S. designed the study and supervised it together with G.B. Data collection, data assessment and statistical analyses were carried out by V.A.S. Data evaluation and discussion of results was a joint effort by all authors (V.A.S., G.B., T.S. and M.T.). V.A.S contributed to the first draft of the manuscript and G.B., T.S. and M.T. improved further versions. All authors gave final approval for publication.
Competing interests. The authors declare no competing interests.
Funding. We acknowledge funding from the European Union's Horizon 2020 research and innovation programme by a Marie Skłodowska-Curie grant (grant agreement no. 722842, ITN Plant-inspired Materials and Surfaces–PlaMatSu) to the authors. The article processing charge was funded by the Baden-Württemberg Ministry of Science, Research and Art and the University of Freiburg in the funding programme Open Access Publishing.
Acknowledgements. We thank the gardeners of the Botanic Garden, University of Freiburg for cultivating the Hevea brasiliensis tree, and the engineers and technicians of the Technical Workshop of the Institute for Biology II/III for the construction of the traction force measurement device. We thank Dr. Tim Kampowski for valuable discussions on the statistical analyses. We also thank Thanuja Kanamarlapudi, wife of the first author, for kindly donating her long Indian hair for beetle experiments.

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
