## [Reviewer comments · Royal Society Open Science]

Review History

RSOS-201319.R0 (Original submission)

Review form: Reviewer 1

Is the manuscript scientifically sound in its present form?

Yes

Are the interpretations and conclusions justified by the results?

Yes

Is the language acceptable?

Yes

Do you have any ethical concerns with this paper?

No

Have you any concerns about statistical analyses in this paper?

No

Recommendation?

Major revision is needed (please make suggestions in comments)

Comments to the Author(s)

In their manuscript, Surapaneni and colleagues investigate the different development stages of the surface morphology of the adaxial side of the leaves of the rubber plant, *Hevea brasiliensis*. The authors identify several stages of development and showed by using confocal laser microscopy of polymer leaf replicas that different leaf stages have different morphological characters. Furthermore, they explored the effect of the different surface morphologies on the resulting traction forces by employing Colorado potato beetles (*Leptinotarsa decemlineata*) as a model species. The results of the manuscript are certainly interesting, yet I found several major concerns about the experimental design and the logic of the paper that need to be improved in a revision. Let me point out my major stumbling points below.

Leaf collection

- The design of the development stages lacks precision and is confusing. Most stages strongly overlap in time and are rather based on visual appearances, which is not a strong enough argument to classify them into different categories. The reviewer is aware that there is not a commonly established system, but please justify this better.

Surface replication

- It is clear that glass is replicated to show the influence of the process on flat surface structures. However, this is not explained in the text. Please adjust.

Ridge dimensions

- The apparent roughness parameters R_c , R_{sk} and R_{sm} need proper definition and introduction.

Discussion

- The authors mention that the colour differences of the various stages are due to change in the surface morphology. There is no evidence for this statement in the manuscript. The authors are encouraged to either show evidence or remove this statement.
- The skewness comparison of the leaf replica surfaces of stage 4 and 5 are is due to the fact of the way the plant was grown. Please elaborate on this idea. At the moment there is not enough data suggesting that "possible additional" cuticular wax deposition is the reason for indifference in skewness.
- The authors explain that in early stage ontogeny, *H. brasiliensis* employ cyanogenic chemical defences. Which defences? And if so, why are they not effective against the litter beetle *Luprops tristis* that has been shown to prefer premature leaves?
- "The fact that the variables of surface chemistry have been eliminated in our study implies that the dimensional changes in micro-scale plant surface structures alone can influence the way that insects forage and feed on plant organs". This is a strong statement and needs proper support, particularly as the chemical aspect has not been investigated properly as only one type of chemical surface has been investigated. Traction forces have been only been performed on PDMS replicas and not on the original surfaces.
- The authors mention that the presented results provide insights for both technical and genetic agricultural solutions. Please elaborate on these solutions.

Figure 6: The graph is losing its potency, the way it is currently displayed. It would be much more efficient to plot the data in percentage and normalized to the glass surface to be quickly intuitive, also given the variations across beetles, I assume.

Figure 4 and the connected results seem to be a specialised presentation of artefacts and as such rather should be placed in the SI than the main text. The logic of the manuscript greatly suffers from this.

Figure 7 seems to be superfluous, as all details are known in literature and could just be referred to.

Minor corrections:

P7L56 replicates replicas

P9L10 the real high aspect ratios could be measured; rephrase

All in all, the presented results are interesting but it seems that the focus of the core findings is lost. This reviewer gets the impression that a lot of data is generated without providing a comprehensive story around it. In addition, the manuscript reads like incoherent, as if several parts have been written by different authors, so please have the manuscript reads again by an English speaker.

Review form: Reviewer 2 (Benjamin Adroit)

Is the manuscript scientifically sound in its present form?

Yes

Are the interpretations and conclusions justified by the results?

Yes

Is the language acceptable?

Yes

Do you have any ethical concerns with this paper?

No

Have you any concerns about statistical analyses in this paper?

No

Recommendation?

Accept with minor revision (please list in comments)

Comments to the Author(s)

Dear Authors,

First of all, congratulation for such study.

The aims are well exposed, the statistics work is well conduct and despite the quantity of data, everything is understandable.

I join the PDF (Appendix A) of your manuscript only with few minor comments which are not affect the study itself.

I have only ONE remark that I would like to share to you. Something that I was a bit "frustrated" to not understand why those specific chose of plant and insect.

Let me explain.

As you know, plants and insects are constantly interacting in complex ways through forest communities since hundreds of millions of years. For both mutualistic (such as pollination) and unilateral (such as herbivory) we are talking about a co-evolution, sometimes very specific between plant and insect species. Consequently, some plants and insects developed some characteristics that are specifics to their interaction.

In your introduction to your study, it looks like you choose one plant and one insect that are not specifically interact to each other and I would like to wonder why?

I full understand that some plant characteristics can be easier to analyse one some specific species, same for insect, maybe the one of this study is easier for you to handle. To me, it's not explicit in the introduction why to choose those species (both plant and insect) and it frustrated me to think that despite this experimentation, which is very interesting, this "interaction" of this insect walking on this leaf is not existing in real life.

This is the only point that I would like authors responds in their introduction in 1-2 sentences, please.

Decision letter (RSOS-201319.R0)

Dear Mr Surapaneni,

The Editors assigned to your paper RSOS-201319 "Spatiotemporal development of cuticular ridges on leaf surfaces of *Hevea brasiliensis* alters insect attachment" have now received comments from reviewers and would like you to revise the paper in accordance with the reviewer comments and any comments from the Editors. Please note this decision does not guarantee eventual acceptance.

Please submit your revised manuscript and required files (see below) no later than 21 days from today's (ie 10-Sep-2020) date. Note: the ScholarOne system will 'lock' if submission of the revision is attempted 21 or more days after the deadline. If you do not think you will be able to meet this deadline please contact the editorial office immediately.

on behalf of the Associate Editor, and Professor Kevin Padian (Subject Editor)
openscience@royalsociety.org

Editor Comments to Author:

Thank you for your submission. On balance the reviewers seem generally happy with it, but there are some comments that may take a while to address. Best wishes for your revision.

Reviewer comments to Author:

Reviewer: 1

Comments to the Author(s)

In their manuscript, Surapaneni and colleagues investigate the different development stages of the surface morphology of the adaxial side of the leaves of the rubber plant, *Hevea brasiliensis*. The authors identify several stages of development and showed by using confocal laser microscopy of polymer leaf replicas that different leaf stages have different morphological characters. Furthermore, they explored the effect of the different surface morphologies on the resulting traction forces by employing Colorado potato beetles (*Leptinotarsa decemlineata*) as a model species. The results of the manuscript are certainly interesting, yet I found several major concerns about the experimental design and the logic of the paper that need to be improved in a revision. Let me point out my major stumbling points below.

Leaf collection

- The design of the development stages lacks precision and is confusing. Most stages strongly overlap in time and are rather based on visual appearances, which is not a strong enough argument to classify them into different categories. The reviewer is aware that there is not a commonly established system, but please justify this better.

Surface replication

- It is clear that glass is replicated to show the influence of the process on flat surface structures. However, this is not explained in the text. Please adjust.

Ridge dimensions

- The apparent roughness parameters R_c , R_{sk} and R_{sm} need proper definition and introduction.

Discussion

- The authors mention that the colour differences of the various stages are due to change in the surface morphology. There is no evidence for this statement in the manuscript. The authors are encouraged to either show evidence or remove this statement.
- The skewness comparison of the leaf replica surfaces of stage 4 and 5 are is due to the fact of the way the plant was grown. Please elaborate on this idea. At the moment there is not enough data suggesting that "possible additional" cuticular wax deposition is the reason for indifference in skewness.
- The authors explain that in early stage ontogeny, *H. brasiliensis* employ cyanogenic chemical defences. Which defences? And if so, why are they not effective against the litter beetle *Luprops tristis* that has been shown to prefer premature leaves?
- "The fact that the variables of surface chemistry have been eliminated in our study implies that the dimensional changes in micro-scale plant surface structures alone can influence the way that insects forage and feed on plant organs". This is a strong statement and needs proper support, particularly as the chemical aspect has not been investigated properly as only one type of chemical surface has been investigated. Traction forces have been only been performed on PDMS replicas and not on the original surfaces.
- The authors mention that the presented results provide insights for both technical and genetic agricultural solutions. Please elaborate on these solutions.

Figure 6: The graph is losing its potency, the way it is currently displayed. It would be much more efficient to plot the data in percentage and normalized to the glass surface to be quickly intuitive, also given the variations across beetles, I assume.

Figure 4 and the connected results seem to be a specialised presentation of artefacts and as such rather should be placed in the SI than the main text. The logic of the manuscript greatly suffers from this.

Figure 7 seems to be superfluous, as all details are known in literature and could just be referred to.

Minor corrections:

P7L56 replicates replicas

P9L10 the real high aspect ratios could be measured; rephrase

All in all, the presented results are interesting but it seems that the focus of the core findings is lost. This reviewer gets the impression that a lot of data is generated without providing a comprehensive story around it. In addition, the manuscript reads like incoherent, as if several parts have been written by different authors, so please have the manuscript read again by an English speaker.

Reviewer: 2

Comments to the Author(s)

Dear Authors,

First of all, congratulation for such study.

The aims are well exposed, the statistics work is well conduct and despite the quantity of data, everything is understandable.

I join the PDF of your manuscript only with few minor comments which are not affect the study itself.

I have only ONE remark that I would like to share to you. Something that I was a bit "frustrated" to not understand why those specific chose of plant and insect. Let me explain.

As you know, plants and insects are constantly interacting in complex ways through forest communities since hundreds of millions of years. For both mutualistic (such as pollination) and unilateral (such as herbivory) we are talking about a co-evolution, sometimes very specific between plant and insect species. Consequently, some plants and insects developed some characteristics that are specific to their interaction.

In your introduction to your study, it looks like you choose one plant and one insect that are not specifically interact to each other and I would like to wonder why?

I full understand that some plant characteristics can be easier to analyse one some specific species, same for insect, maybe the one of this study is easier for you to handle. To me, it's not explicit in the introduction why to choose those species (both plant and insect) and it frustrated me to think that despite this experimentation, which is very interesting, this "interaction" of this insect walking on this leaf is not existing in real life.

This is the only point that I would like authors responds in their introduction in 1-2 sentences, please.

===PREPARING YOUR MANUSCRIPT===

- one version identifying all the changes that have been made (for instance, in coloured highlight, in bold text, or tracked changes);
- a 'clean' version of the new manuscript that incorporates the changes made, but does not highlight them.

This version will be used for typesetting if your manuscript is accepted.

Please ensure that you include an acknowledgements' section before your reference list/bibliography. This should acknowledge anyone who assisted with your work, but does not

qualify as an author per the guidelines at <https://royalsociety.org/journals/ethics-policies/openness/>.

===PREPARING YOUR REVISION IN SCHOLARONE===

<https://royalsociety.org/journals/authors/author-guidelines/#data>. You should ensure that

you cite the dataset in your reference list. If you have deposited data etc in the Dryad repository, please include both the 'For publication' link and 'For review' link at this stage.

Author's Response to Decision Letter for (RSOS-201319.R0)

See Appendix B.

RSOS-201319.R1 (Revision)

Review form: Reviewer 1

Is the manuscript scientifically sound in its present form?

Yes

Are the interpretations and conclusions justified by the results?

Yes

Is the language acceptable?

Yes

Do you have any ethical concerns with this paper?

No

Have you any concerns about statistical analyses in this paper?

No

Recommendation?

Accept with minor revision (please list in comments)

Comments to the Author(s)

This revision has much improved and I see no major obstacles in the way. Only a few small things: i) make sure that all Latin names are italics, also in the titles of the references. ii) Same for the parameters in the main text, R_c etc, which should also be italics.

Otherwise this is a nice contribution that I look forward to seeing in print!

Decision letter (RSOS-201319.R1)

Dear Mr Surapaneni

On behalf of the Editors, we are pleased to inform you that your Manuscript RSOS-201319.R1 "Spatiotemporal development of cuticular ridges on leaf surfaces of *Hevea brasiliensis* alters insect attachment" has been accepted for publication in Royal Society Open Science subject to minor revision in accordance with the referees' reports. Please find the referees' comments along with any feedback from the Editors below my signature.

When submitting your revised paper, please ensure to update the following email address, which is currently marked as invalid by our system:

- georg.bold@biologie.uni-freiburg.de

Please submit your revised manuscript and required files (see below) no later than 7 days from today's (ie 29-Sep-2020) date. Note: the ScholarOne system will 'lock' if submission of the revision is attempted 7 or more days after the deadline. If you do not think you will be able to meet this deadline please contact the editorial office immediately.

on behalf of the Associate Editor and Professor Kevin Padian (Subject Editor)
openscience@royalsociety.org

Associate Editor Comments to Author:

A few minor typographical changes to make, but otherwise you're all set - well done!

Reviewer comments to Author:

Reviewer: 1

Comments to the Author(s)

This revision has much improved and I see no major obstacles in the way. Only a few small things: i) make sure that all Latin names are italics, also in the titles of the references. ii) Same for the parameters in the main text, R_c etc, which should also be italics.

Otherwise this is a nice contribution that I look forward to seeing in print!

===PREPARING YOUR MANUSCRIPT===

- one version identifying all the changes that have been made (for instance, in coloured highlight, in bold text, or tracked changes);
- a 'clean' version of the new manuscript that incorporates the changes made, but does not highlight them. This version will be used for typesetting.

===PREPARING YOUR REVISION IN SCHOLARONE===

Author's Response to Decision Letter for (RSOS-201319.R1)

See Appendix C.

Decision letter (RSOS-201319.R2)

Dear Mr Surapaneni,

It is a pleasure to accept your manuscript entitled "Spatiotemporal development of cuticular ridges on leaf surfaces of *Hevea brasiliensis* alters insect attachment" in its current form for publication in Royal Society Open Science.

on behalf of Prof Kevin Padian (Subject Editor)
openscience@royalsociety.org

Appendix A**ROYAL SOCIETY
OPEN SCIENCE****Spatiotemporal development of cuticular ridges on leaf
surfaces of *Hevea brasiliensis* alters insect attachment**

Journal:	Royal Society Open Science
Manuscript ID	RSOS-201319
Article Type:	Research
Date Submitted by the Author:	29-Jul-2020
Complete List of Authors:	Surapaneni, Venkata; Albert-Ludwigs-Universitat Freiburg Fakultat fur Biologie, Faculty of Biology Bold, Georg; Albert-Ludwigs-Universitat Freiburg Fakultat fur Biologie, Faculty of Biology Speck, Thomas; Albert-Ludwigs-Universitat Freiburg, Plant Biomechanics Group, Botanic Garden University of Freiburg Thielen, Marc; Albert-Ludwigs-Universitat Freiburg, Institute for Biology, Botanic Garden
Subject:	ecology < BIOLOGY, evolution < BIOLOGY, biomechanics < BIOLOGY
Keywords:	ridge, cuticle, ontogeny, plant defence, plant-insect interaction, adhesion
Subject Category:	Organismal and Evolutionary Biology

Author-supplied statements

Relevant information will appear here if provided.

Ethics

Does your article include research that required ethical approval or permits?:

This article does not present research with ethical considerations

Statement (if applicable):

CUST_IF_YES_ETHICS :No data available.

Data

It is a condition of publication that data, code and materials supporting your paper are made publicly available. Does your paper present new data?:

Yes

Statement (if applicable):

The datasets supporting the results presented in this article are uploaded and available online as .zip files at: <https://freidok.uni-freiburg.de/data/166607> with DOI: 10.6094/UNIFR/166607 [60]

Conflict of interest

I/We declare we have no competing interests

Statement (if applicable):

CUST_STATE_CONFLICT :No data available.

Authors' contributions

This paper has multiple authors and our individual contributions were as below

Statement (if applicable):

M.T and T.S designed the study and supervised it together with G.B. Data collection, data assessment and statistical analyses were carried out by V.A.S. Data evaluation and discussion of results was a joint effort by all authors (V.A.S., G.B., T.S. and M.T.). V.A.S contributed to the first draft of the manuscript and G.B., T.S. and M.T. improved further versions. All authors gave final approval for publication.

**Spatiotemporal development of cuticular ridges on leaf surfaces of *Hevea brasiliensis* alters**
**insect attachment**

Venkata A. Surapaneni^{*1}, Georg Bold², Thomas Speck³, Marc Thielen⁴

10 ^{1,2,3,4}*Plant Biomechanics Group, Botanic Garden, Faculty of Biology, University of Freiburg,*
*Schänzlestrasse 1, 79104 Freiburg, Germany.*

14 ^{1,2,3,4}*FIT, Freiburg Center for Interactive Materials and Bioinspired Technologies, Georges-*
15 *Köhler-Allee 105, 79110 Freiburg, Germany.*

18 ^{1,2,3,4}*FMF, Freiburg Materials Research Center, Stefan-Meier-Strasse 21, 79104 Freiburg,*
*Germany.*

³*Cluster of Excellence livMatS@ FIT- Freiburg Center for Interactive Materials and Bioinspired*
*Technologies, University of Freiburg, Georges-Köhler-Allee 105, 79110 Freiburg, Germany.*

28 ^{*1}*amarnadh.sv@bio.uni-freiburg.de (Corresponding author)*

²*georg.bold@biologie.uni-freiburg.de*

³*thomas.speck@biologie.uni-freiburg.de*

⁴*marc.thielen@biologie.uni-freiburg.de*

Abstract

[revised manuscript text omitted]

replicas. The traction forces on flat PDMS glass replica surfaces were much lower than on glass
surfaces ($p < 0.001, n=40$) showing the effect of surface chemistry on insect adhesion. Compared
with the flat PDMS surfaces, the replicas of leaves at S1 ($p = 0.996, n=40$) and S2A ($p = 0.963,$

$n=40$) did not differ significantly in insect traction forces. Even though the dimensions of the
ridges were influenced by the plant remnants at S2B (replicas from tainted moulds), the traction
forces showed a decrease in magnitude compared with those on the surfaces towards the apex
(S2A). In reality, the insect traction forces on the leaves at stages S2B and S3 might even be lower
than or comparable with the traction forces on leaves at stage S4, because of the critical roughness
(0.3 - 3 μm) of the cuticular ridges at the transition stages. The traction forces on the adult leaf
replicas (S4 and S5) were significantly lower ($p < 0.001$) compared with those on young leaves
(S1 and S2A). The mean traction forces at stage S5 increased significantly ($p = 0.049$, $n=40$)
compared with the forces at stage S4. An analysis of random effects showed the relatively lower
influence of replicates, i.e. the variation within leaves in each stage (4.2%, variance: 0.003, residual
variance: 0.038), compared with that from the beetles (27.4%, variance: 0.014, residual variance:
0.038) and the differences in traction forces within replicates in each stage were not significant.
During the experiments, the interaction of beetles with the protruding veins was unavoidable.
Nonetheless, these structures did not seem to affect the reduction of traction forces during leaf
growth. The log-transformed mean of the traction force and mean aspect ratio (AR) values taken
over replicates showed a strong association (*Pearson's product-moment correlation*, $R = -0.91$, df
$= 28$, $p < 1.9e-12$) (Supplementary Fig. S5). The association between the log-transformed values
of the traction forces was also strong compared with R_c and R_a (both, $R = -0.90$, $df = 28$, $p < 1.0e-$
11). However, no correlation was found between log-transformed values of traction forces and
R_{sm} ($R = 0.28$, $df = 28$, $p = 0.18$).

37 38 **4. Discussion**

In this study, we defined the leaf growth stages of *Hevea brasiliensis* by means of a combination
of visual colour changes on the adaxial side of the leaves (S1, S2 and S3) and growth (S4 and S5).
Initial screening using confocal laser scanning microscopy helped to verify the segregation of the
stages S1, S2 and S3 based on the colour differences as they clearly corresponded to the drastic
changes in the surface microstructure. This differed from the definition of leaf growth stages for
the same species in earlier studies, which only involved macroscale morphological parameters or
physiological data. Based on morphology, for example, Dijkman, 1951 [41] discerned four leaf-
growth stages A, B, C and D corresponding to bud burst, young, fully expanded and mature states,
respectively, with stages A to C acting as sink leaves having almost no lignin and varying
physiologically, for example, in chlorophyll amount and cyanogenic capacity [42, 43], and with
stage D leaves being mature source leaves having a high amount of lignin [44, 45]. As our work
was aimed at understanding the effect of changes in the morphology of surface microstructures on

insect walking forces, we based our classification on these microstructures, which, nevertheless,
were also reflected to a large extent by macroscopically visible features (i.e. colour). These
microscopic changes on the leaf surfaces however did not match with the previously defined stages
by Dijkman, 1951 [41]. The growth stages S1, S4 and S5 defined in our study corresponded to the
stages B, C and D, respectively. From this comparison, the leaves from stages A to B contained
smooth epidermal cells and the cuticular ridges developed completely until stage C. The ridge

[revised manuscript text omitted]

- 26. Perassadko AG, Gorb SN. 2004. Proceedings of the conference, Bionik-2004 22-23, 257-263,
- April 2004, Hanover, Germany.
- 27. Voigt D, Schuppert JM, Dattinger S, Gorb SN. 2008. Sexual dimorphism in the attachment
- ability of the Colorado potato beetle *Leptinotarsa decemlineata* (Coleoptera: Chrysomelidae)
- to rough substrates. *J. Insect Physiol.* **54**, 765-776.
- <https://doi.org/10.1016/j.jinsphys.2008.02.006>
- 28. Al Bitar L, Voigt D, Zebitz CP, Gorb SN. 2010. Attachment ability of the codling moth *Cydia*
- *pomonella* L. to rough substrates. *J. Insect Physiol.* **56**, 1966-72.
- <https://doi.org/10.1016/j.jinsphys.2010.08.021>
- 29. Bullock JMR, Federle W. 2011. The effect of surface roughness on claw and adhesive hair
- performance in the dock beetle *Gastrophysa viridula*. *Insect Sci.* **18**, 298–304.
- <https://doi.org/10.1111/j.1744-7917.2010.01369.x>
- 30. Wolff JO, Gorb SN. 2012. Surface roughness effects on attachment ability of the spider
- *Philodromus dispar* (Araneae, Philodromidae). *J. Exp. Biol.* **215**, 179-184.
- <https://doi.org/10.1242/jeb.061507>
- 31. Priyadarshan PM. 2017. *Biology of Hevea Rubber*. Springer International Publishing AG,
- Cham, Switzerland. ISBN 978-3-319-54506-6. <https://doi.org/10.1007/978-3-319-54506-6>
- 32. Rao AN. 1963. Reticulate cuticle on leaf epidermis in *Hevea brasiliensis* Muell. *Nature* **197**,
- 1125-1126. <https://doi.org/10.1038/1971125b0>
- 33. Gorb SN. 2001. *Attachment Devices of the Insect Cuticle*, Kluwer Academic Publishers,
- London, United Kingdom. ISBN 0-7923-7153-4. <https://doi.org/10.1007/0-306-47515-4>
- 34. Kumar C, Le Houérou V, Speck T, Bohn HF. 2018. Straightforward and precise approach to
- replicate complex hierarchical structures from plant surfaces onto soft matter polymer. *R. Soc.*
- *Open Sci.* **5**, 172132. <https://doi.org/10.1098/rsos.172132>
- 35. Leach RK. 2010. *Fundamental Principles of Engineering Nanometrology*. Elsevier Inc.,
- Oxford, United Kingdom. ISBN–13: 978-0-08-096454-6. [https://doi.org/10.1016/c2009-0-](https://doi.org/10.1016/c2009-0-20339-4)
- [20339-4](https://doi.org/10.1016/c2009-0-20339-4)

36. Schindelin J, Arganda-Carreras I, Frise E et al. 2012. Fiji: an open-source platform for
biological-image analysis. *Nat. Methods* **9**, 676–682. <https://doi.org/10.1038/nmeth.2019>
37. Neinhuis C, Edelmann HG. 1996. Methanol as a rapid fixative for the investigation of plant
surfaces by SEM. *J. Microsc.* **184**, 14-16. <https://doi.org/10.1046/j.1365-2818.1996.d01-110.x>
38. R Core Team. 2019. R: A Language and Environment for Statistical Computing, R Foundation
for Statistical Computing, Vienna, Austria, <https://www.R-project.org>.
39. Bates D, Maechler M, Bolker B, Walker S. 2015. Fitting Linear Mixed-Effects Models Using
lme4. *J. Stat. Softw.* **67**, 1-48.
40. Lenth R. 2019. emmeans: Estimated Marginal Means, aka Least-Squares Means. R package
version 1.4.2
41. Dijkman MJ. 1951. Hevea: thirty years of research in the Far East. Coral Gables, FL:
University of Miami Press, 329.
42. Lieberei R, Fock HP, Biehl B. 1996. Cyanogenesis inhibits active pathogen defense in plants:
Inhibition by gaseous HCN of photosynthetic CO₂-fixation and respiration in intact leaves.
*Angew. Bot.* **70**, 230–238.
43. Lieberei R. 1984. Cyanogenese und Resistenz. Habilitationsschrift, Naturwissenschaftliche
Fakultät, Technische Universität Braunschweig, Germany, 265.
44. Voß K. 2001. Biologische Bedeutung und Aktivierbarkeit der b-D-Glycosidase in Blättern von
Hevea brasiliensis (Willd.) Muell. Arg. (1865). PhD thesis, University of Hamburg, Germany.
45. Lieberei R. 2007. South American Leaf Blight of the Rubber Tree (Hevea spp.): New Steps in
Plant Domestication using Physiological Features and Molecular Markers. *Ann. Bot.* **100**,
1125–1142. <https://doi.org/10.1093/aob/mcm133>
46. Fang Y, Mei H, Zhou B, Xiao X, Yang M, Huang Y, Long X, Hu S, Tang C. 2016. De novo
transcriptome analysis reveals distinct defense mechanisms by young and mature leaves
of *Hevea brasiliensis* (Para Rubber Tree). *Sci. Rep.* **6**, 33151.
<https://doi.org/10.1038/srep33151>
47. Hong L, Brown J, Segerson NA, Rose JKC, Roeder AHK. 2017. CUTIN SYNTHASE 2
Maintains Progressively Developing Cuticular Ridges in Arabidopsis Sepals. *Mol. Plant* **10**,
560–574. <https://doi.org/10.1016/j.molp.2017.01.002>
48. Martens P. 1933. Recherches sur la cuticule. *Protoplasma* **20**, 483–515.
<https://doi.org/10.1007/bf02674844>

49. Kourouniotti RLA, Band LR, Fozard JA, Hampstead A, Lovrics A, Moyroud E, Vignolini S,
King JR, Jensen OE, Glover BJ. 2013. Buckling as an origin of ordered cuticular patterns in
flower petals. *J. R. Soc. Interface* **10**, 20120847. <https://doi.org/10.1098/rsif.2012.0847>
50. Rhee Y, Hlousek-Radojicic A, Ponsamuel J, Liu D, Post-Beittenmiller D. 1998. Epicuticular
Wax Accumulation and Fatty Acid Elongation Activities Are Induced during Leaf
Development of Leeks. *Plant Physiol.* **116**, 901-911. <https://doi.org/10.1104/pp.116.3.901>
51. Hauke V, Schreiber L. 1998. Ontogenetic and seasonal development of wax composition and
cuticular transpiration of ivy (*Hedera helix* L.) sun and shade leaves. *Planta* **207**, 67–75.
<https://doi.org/10.1007/s004250050456>
52. Bringe K, Schumacher CFA, Schmitz-Eiberger M, Steiner U, Oerke EC. 2006. Ontogenetic
variation in chemical and physical characteristics of adaxial apple leaf surfaces.
*Phytochemistry* **67**, 161-170. <https://doi.org/10.1016/j.phytochem.2005.10.018>
53. Holloway PJ. 1970. Surface factors affecting the wetting of leaves. *Pestic. Sci.* **1**, 156-163.
<https://doi.org/10.1002/ps.2780010411>
54. Jayasinghe CK. 1999. Pests and diseases of hevea rubber and their geographical distribution.
*Bulletin of the Rubber Research Institute of Sri Lanka* **40**, 1-8.
55. Sabu TK, Vinod KV. 2009a. Food Preferences of the Rubber Plantation Litter Beetle, *Luprops*
*tristis*, a Nuisance Pest in Rubber Tree Plantations. *J. Insect Sci.* **9**, 72-77.
<https://doi.org/10.1673/031.009.7201>
56. Sabu TK, Vinod KV. 2009b. Population dynamics of the rubber plantation litter beetle *Luprops*
*tristis*, in relation to annual cycle of foliage phenology of its host, the para rubber tree, *Hevea*
*brasiliensis*. *J. Insect Sci.* **9**, 56-66. <https://doi.org/10.1673/031.009.5601>
57. Lieberei R. 1986. Cyanogenesis of *Hevea brasiliensis* during infection with *Microcyclus ulei*.
*J. Phytopathol.* **115**, 134–146. <https://doi.org/10.1111/j.1439-0434.1986.tb00870.x>
58. Barton KE, Edwards KF, Koricheva J. 2019. Shifts in woody plant defence syndromes during
leaf development. *Funct. Ecol.* **33**, 2095– 2104. <https://doi.org/10.1111/1365-2435.13435>
59. Aharoni A, Dixita S, Jetterb R, Thoenesa E, van Arkela G, Pereira A. 2004. The SHINE clade
of AP2 domain transcription factors activates wax biosynthesis, alters cuticle properties, and
confers drought tolerance when overexpressed in Arabidopsis. *Plant Cell* **16**, 2463– 2480.
<https://doi.org/10.1105/tpc.104.022897>

60. Surapaneni VA, Bold G, Thielen M, Speck T. 2020. Raw data and code for "Spatiotemporal
development of cuticular ridges on leaf surfaces of *Hevea brasiliensis* alters insect attachment".
version 1. *FreiDok Plus, UB Freiburg*. <https://doi.org/10.6094/UNIFR/166607>

Acknowledgements

We thank the gardeners of the Botanic Garden, University of Freiburg for cultivating the *Hevea brasiliensis* tree, and the engineers and technicians of the Technical Workshop of the Institute for Biology II/III for the construction of the traction force measurement device. We thank Dr. Tim Kampowski for valuable discussions on the statistical analyses. We also thank Thanuja Kanamarlapudi, wife of the first author, for kindly donating her long Indian hair for beetle experiments.

Author contributions

M.T and T.S designed the study and supervised it together with G.B. Data collection, data assessment and statistical analyses were carried out by V.A.S. Data evaluation and discussion of results was a joint effort by all authors (V.A.S., G.B., T.S. and M.T.). V.A.S contributed to the first draft of the manuscript and G.B., T.S. and M.T. improved further versions. All authors gave final approval for publication.

Competing interests

The authors declare no competing interests.

Funding

We acknowledge funding from the European Union's Horizon 2020 research and innovation programme by a Marie Skłodowska-Curie grant (grant agreement No. 722842, ITN Plant-inspired Materials and Surfaces—PlaMatSu) to the authors.

Data accessibility

The datasets supporting the results presented in this article are uploaded and available online as .zip files at: <https://freidok.uni-freiburg.de/data/166607> with DOI: 10.6094/UNIFR/166607 [60]

Figure legends

Fig. 1: Leaf transition: Leaves from the tip of a branch of *Hevea brasiliensis* tree (height > 10m) with almost vertically drooping leaves during young stages; the leaves gradually position themselves more horizontally as they grow. The image shows (a) young leaves at stage S1, (b) leaves in transition to stage S2 with the surface progressing from shiny brown to dull pale green acropetally, (c) leaves at stage S3 and (d) adult leaves (S4 and S5).

Fig. 2: Development of leaf and of cuticular ridges (a) *Hevea brasiliensis* leaves at various growth stages with a colour spectrum (SpyderCHECKR 24 - SCK200, Datacolor AG Europe, Rotkreuz, Switzerland). (b-g) Corresponding CLSM images of the leaf replicas at various stages: (b) Stage 1, (c) Stage 2A, (d) Stage 2B, (e) Stage 3, (f) Stage 4 and (g) Stage 5. The appearance of the pale green colour towards the basal region of leaf at stage 2B coincides with the appearance of microscale ridges. Both the colour and the ridges progress acropetally and further develop during the following growth stages.

Fig. 3: Leaf replication Schematic representation of original leaves and their negative (epoxy) and positive (PDMS) replicas. A large portion of the epoxy replicas of stages S2B (basal region of the stage 2 leaves) and S3 was fused with leaf remnants. A few patches and the region close to the base of the leaves at stages S2B and S3 were free of leaf remnants. The surfaces of the PDMS positive replicas did not have any leaf remnants.

Fig. 4: Remains of plant material on replicas Except for a few patches (as shown in Fig. 2 and Fig. 3), the entire area of the epoxy replicas of leaves at stages S2B and S3 retained plant cuticular material, even after KOH treatment. (a - b) CLSM images of positive replicas of leaves at stages (a) S2B and (b) S3. (c) CLSM image of a region much closer to the base of the leaf replica at transition stage S3 in which the ridge morphology is much more similar to that of adult stages (Fig. 3).

Fig. 5: Ridge aspect ratio vs. leaf stage Boxplot showing variation in the ridge (roughness) aspect ratio for glass, flat PDMS and replicas of leaves at various leaf stages. For stages S2B and S3, realistic aspect ratio values are given for clean replicas from tiny patches and for the replicas from contaminated moulds representing the remaining surface, which were used for traction experiments (box plots in grey; they underestimate the aspect ratio) (see Results section).

Fig. 6: Insect traction forces Box plot showing the differences in traction forces of *Leptinotarsa decemlineata* ($n=40$) for a set of statistical replicates of PDMS replica surfaces of leaves at various growth stages compared with glass and flat PDMS glass replicas. The traction force values for

stage 2B and stage 3 could only be calculated from replicas from contaminated moulds (box plots
in grey) and therefore overestimate the real values (see Results section).

**Fig. 7: Insect adhesion in interaction with plant surface:** (a) Colorado potato beetle (*Leptinotarsa*
*decemlineata*) walking on a leaf replica, (b) SEM image showing tarsal features on the mid leg of
female Colorado potato beetle *Leptinotarsa decemlineata*, (c) SEM image showing spatula-type
terminal ends of the hairy attachment system of the insect. (d - e) Schematic representation of the
interfacial contact area formation between the terminal ends of insect setae and the leaf surfaces
at early (d) and adult (e) growth stages of *Hevea brasiliensis* leaves.

**Tables**

**Table 1. Median values of the roughness parameters from all the replicates of glass, flat**
**PDMS and leaf replica surfaces. For stages S2B and S3, values from clean small areas are**
**given representing the realistic high aspect ratio of ridges (values not in parentheses);**
**values from replicas from contaminated moulds underestimating the real aspect ratio**
**(values in parentheses) are given that have been used in traction experiments of these two**
**stages.**

Surface	Age (days)	Ra (μm)	Rc (μm)	Rsm (μm)	Aspect ratio = Rc/Rsm	Rsk
Glass	-	0.0004	0.001	1.020	0.001	0.346
Flat PDMS	-	0.0005	0.002	1.250	0.002	0.328
S1	13 ± 2	0.008	0.031	4.465	0.006	-0.906
S2A	15 ± 3	0.010	0.038	4.440	0.008	-0.768
S2B	15 ± 3	0.111 (0.044)	0.358 (0.152)	1.095 (1.465)	0.323 (0.102)	-0.265 (-0.48)
S3	16 ± 3	0.132 (0.053)	0.418 (0.173)	1.13 (1.33)	0.378 (0.129)	-0.27 (-0.495)
S4	21 ± 4	0.181	0.542	2.105	0.261	0.215
S5	> 60	0.190	0.579	2.020	0.283	0.035

Figures

Fig. 1: Leaf transition: Leaves from the tip of a branch of *Hevea brasiliensis* tree (height > 10m) with almost vertically drooping leaves during young stages; the leaves gradually position themselves more horizontally as they grow. The image shows (a) young leaves at stage S1, (b) leaves in transition to stage S2 with the surface progressing from shiny brown to dull pale green acropetally, (c) leaves at stage S3 and (d) adult leaves (S4 and S5).

Fig. 2: Development of leaf and of cuticular ridges (a) *Hevea brasiliensis* leaves at various growth stages with a colour spectrum (SpyderCHECKR 24 - SCK200, Datacolor AG Europe, Rotkreuz, Switzerland). (b-g) Corresponding CLSM images of the leaf replicas at various stages: (b) Stage 1, (c) Stage 2A, (d) Stage 2B, (e) Stage 3, (f) Stage 4 and (g) Stage 5. The appearance of the pale green colour towards the basal region of leaf at stage 2B coincides with the appearance of microscale ridges. Both the colour and the ridges progress acropetally and further develop during the following growth stages.

Fig. 3: Leaf replication Schematic representation of original leaves and their negative (epoxy) and positive (PDMS) replicas. A large portion of the epoxy replicas of stages S2B (basal region of the stage 2 leaves) and S3 was fused with leaf remnants. A few patches and the region close to the base of the leaves at stages S2B and S3 were free of leaf remnants. The surfaces of the PDMS positive replicas did not have any leaf remnants.

Fig. 4: *Remains of plant material on replicas* Except for a few patches (as shown in Fig. 2 and Fig. 3), the entire area of the epoxy replicas of leaves at stages S2B and S3 retained plant cuticular material, even after KOH treatment. (a - b) CLSM images of positive replicas of leaves at stages (a) S2B and (b) S3. (c) CLSM image of a region much closer to the base of the leaf replica at transition stage S3 in which the ridge morphology is much more similar to that of adult stages (Fig. 3).

Fig. 5: Ridge aspect ratio vs. leaf stage Boxplot showing variation in the ridge (roughness) aspect ratio for glass, flat PDMS and replicas of leaves at various leaf stages. For stages S2B and S3, realistic aspect ratio values are given for clean replicas from tiny patches and for the replicas from contaminated moulds representing the remaining surface, which were used for traction experiments (box plots in grey; they underestimate the aspect ratio) (see Results section).

Fig. 6: Insect traction forces Box plot showing the differences in traction forces of *Leptinotarsa decemlineata* ($n=40$) for a set of statistical replicates of PDMS replica surfaces of leaves at various growth stages compared with glass and flat PDMS glass replicas. The traction force values for stage 2B and stage 3 could only be calculated from replicas from contaminated moulds (box plots in grey) and therefore overestimate the real values (see Results section).

Fig. 7: Insect adhesion in interaction with plant surface: (a) Colorado potato beetle (*Leptinotarsa decemlineata*) walking on a leaf replica, (b) SEM image showing tarsal features on the mid leg of female Colorado potato beetle *Leptinotarsa decemlineata*, (c) SEM image showing spatula-type terminal ends of the hairy attachment system of the insect. (d - e) Schematic representation of the interfacial contact area formation between the terminal ends of insect setae and the leaf surfaces at early (d) and adult (e) growth stages of *Hevea brasiliensis* leaves.

Appendix B

Response to editor's and reviewers' comments

The comments are in *black italic font* and our responses are in normal blue font. Any sentences taken from the main manuscript are in green font.

Editor Comments to Author:

Thank you for your submission. On balance the reviewers seem generally happy with it, but there are some comments that may take a while to address. Best wishes for your revision.

We thank the editor for considering our submission. We appreciate that the reviewers are happy with the manuscript. We addressed all comments of the reviewers below, and incorporated the changes in the manuscript. We hope that the editor and the reviewers will find our responses satisfactory.

Reviewer comments to Author:

Reviewer: 1

Comments to the Author(s)

*In their manuscript, Surapaneni and colleagues investigate the different development stages of the surface morphology of the adaxial side of the leaves of the rubber plant, *Hevea brasiliensis*. The authors identify several stages of development and showed by using confocal laser microscopy of polymer leaf replicas that different leaf stages have different morphological characters. Furthermore, they explored the effect of the different surface morphologies on the resulting traction forces by employing Colorado potato beetles (*Leptinotarsa decemlineata*) as a model species. The results of the manuscript are certainly interesting, yet I found several major concerns about the experimental design and the logic of the paper that need to be improved in a revision. Let me point out my major stumbling points below.*

We thank the reviewer for the in-depth review of our manuscript and for finding our results interesting. We appreciate the concerns of the reviewer, and we addressed the reviewer's comments in details below, and incorporated the changes in the revised manuscript wherever appropriate.

Leaf collection

• The design of the development stages lacks precision and is confusing. Most stages strongly overlap in time and are rather based on visual appearances, which is not a strong enough argument to classify them into different categories. The reviewer is aware that there is not a commonly established system, but please justify this better.

As pointed out by the reviewer "...there is not a commonly established system..." to "...design of the development stages" of leaves, which represents notoriously complicated problem. We are sorry for the confusion and we have made changes accordingly in the revised manuscript (in P5-L57). The classification of the leaf stages in this study is based on morphological changes occurring on the surfaces of the leaves and the age of the leaves. We conducted preliminary trial experiments under a confocal laser scanning microscope to analyse whether the visual appearances of the leaves coincided

with the microscale morphological changes. In most stages (stages S1, S2A, S2B, S3), the visual appearances of the leaves notably correlated to the changes in the morphology of the leaf surfaces, which is also an important result in our study. This coincidence of colour and morphology greatly helped us in collecting the leaves at those particular stages for experiments.

We agree that few stages are very close and overlap in time. But, we support our classification based on the observation that strong changes in the morphology of *Hevea brasiliensis* leaf surfaces occur in those stages. Our classification proposed in this study is purely based on the development of the leaves, and therefore, it is – at least in our opinion – valid. We have also compared our classification with previous definitions in the discussion section. We think that with the above additions and clarifications our classification is now better comprehensible.

Surface replication

- *It is clear that glass is replicated to show the influence of the process on flat surface structures. However, this is not explained in the text. Please adjust.*

We did not fully understand what the reviewer exactly meant by “process”. We are assuming that the reviewer was referring to the “replication process”. We have already clarified in our manuscript that glass serves as a flat/unstructured surface and that we used replicas of glass (i.e., PDMS glass replica) to assess the effect of surface chemistry. That is: the surface topology remains the same between glass and PDMS glass replica while the surface chemistry is changed in order to allow comparison with leaf replicas made from the same PDMS material.

Ridge dimensions

- *The apparent roughness parameters R_c , R_{sk} and R_{sm} need proper definition and introduction.*

We thank the reviewer for suggesting this addition. The definitions are included in the revised manuscript.

Discussion

- *The authors mention that the colour differences of the various stages are due to change in the surface morphology. There is no evidence for this statement in the manuscript. The authors are encouraged to either show evidence or remove this statement.*

In our study, we only found that the colour variations of the leaf surfaces coincided with the morphological changes on the leaf surfaces. We mentioned the same in the discussion section of the manuscript – “Initial screening using confocal laser scanning microscopy helped to verify the segregation of the stages S1, S2 and S3 based on the colour differences as they clearly corresponded to the drastic changes in the surface microstructure.” ... “we based our classification on these microstructures, which, nevertheless, were also reflected to a large extent by macroscopically visible features (i.e. colour).”

But, we did not find that the colour differences are due to the changes in the leaf surface morphology. To avoid confusion, we changed the wording – “clearly corresponded to” to “coincided with” in the revised manuscript.

- *The skewness comparison of the leaf replica surfaces of stage 4 and 5 are is due to the fact of the way the plant was grown. Please elaborate on this idea. At the moment there is not enough data suggesting that “possible additional” cuticular wax deposition is the reason for indifference in skewness.*

We thank the reviewer for the very helpful suggestion. We have elaborated the discussion on skewness comparison and removed the sentence on “possible additional” cuticular wax deposition in the revised manuscript.

- *The authors explain that in early stage ontogeny, H. brasiliensis employ cyanogenic chemical defences. Which defences? And if so, why are they not effective against the litter beetle Luprops tristis that has been shown to prefer premature leaves?*

We thank the reviewer for the observation. The young leaves of *H. brasiliensis* release Hydrogen cyanide (HCN) as a response to tissue damage. We added this information in the revised manuscript.

In *H. brasiliensis*, HCN is released by the mixing of cyanogenic precursor – β -glycosides and β -glycosidases [Lieberei R. 2007, ref. 45 in the manuscript]. It seems that some insect herbivores developed the ability to avoid the mixing of these components and therefore to prevent the release of HCN, by various mechanisms [Pentzold et al., 2014]. It may be possible that the beetle *Luprops tristis* also has these abilities. Moreover, it was found that *Luprops tristis* feed only on fallen tender leaves [Sabu and Vinod, 2009a, ref. 55 in the manuscript], where the cyanogenesis could be low [Selmar 1993; Gleadow and Woodrow, 2000]. While this scientific detail is clearly out of scope in the current study, at this moment, we are unaware of any study that specifically answers why the cyanogenic chemical defences of young *H. brasiliensis* leaves are not effective against *Luprops tristis*.

References:

Lieberei R. 2007. South American Leaf Blight of the Rubber Tree (*Hevea spp.*): New Steps in Plant Domestication using Physiological Features and Molecular Markers. *Ann. Bot.* 100, 1125–1142. <https://doi.org/10.1093/aob/mcm133>

Pentzold S, Zagrobelny M, Roelsgaard PS, Møller BL, Bak S. 2014. The Multiple Strategies of an Insect Herbivore to Overcome Plant Cyanogenic Glucoside Defence. *PLoS ONE* 9, 3, e91337. <https://doi.org/10.1371/journal.pone.0091337>

Sabu TK, Vinod KV. 2009a. Food Preferences of the Rubber Plantation Litter Beetle, *Luprops tristis*, a Nuisance Pest in Rubber Tree Plantations. *J. Insect Sci.* 9, 72-77. <https://doi.org/10.1673/031.009.7201>

Gleadow RM, Woodrow IE. 2000. Temporal and spatial variation in cyanogenic glycosides in *Eucalyptus cladocalyx*, *Tree Physiology*, 20, 9, 591–598. <https://doi.org/10.1093/treephys/20.9.591>

Selmar D. 1993. Transport of cyanogenic glucosides: linustatin uptake by *Hevea* cotyledons. *Planta*, 191, 191–199. <https://doi.org/10.1007/BF00199749>

- *“The fact that the variables of surface chemistry have been eliminated in our study implies that the dimensional changes in micro-scale plant surface structures alone can influence the way that insects forage and feed on plant organs”. This is a strong statement and needs proper support, particularly as the chemical aspect has not been investigated properly as only one type of chemical surface has been investigated. Traction forces have been only been performed on PDMS replicas and not on the original surfaces.*

We agree with the reviewer that mentioned sentence is a strong statement. However, this is a major result from our study. For original leaf surfaces, the surface chemistry might change with ontogeny and might affect the insect traction forces. In our study, we aimed at eliminating this variable of surface

chemistry so that the sole effect of the micro-scale morphology of the leaf surfaces can be measured. For this reason, thanks to the reviewer's suggestion, we have modified Fig. 5 and included the results of traction forces on PDMS leaf replicas relative to the traction forces on PDMS glass replicas in the revised manuscript. For comparison, we have also shown the effect of surface chemistry from the traction forces between glass and PDMS glass replicas in Fig. 5 in the revised manuscript. Moreover, traction experiments on original leaf surfaces and other polymer replicas have already been reported in earlier studies [Prüm et al., 2012a; Prüm et al., 2012a; Prüm et al., 2013; refs. 20, 21, 22 in the manuscript], which should serve as a reference.

- *The authors mention that the presented results provide insights for both technical and genetic agricultural solutions. Please elaborate on these solutions.*

We thank the reviewer for the suggestion. In the revised manuscript, we have elaborated on the mentioned solutions.

Figure 6: The graph is losing its potency, the way it is currently displayed. It would be much more efficient to plot the data in percentage and normalized to the glass surface to be quickly intuitive, also given the variations across beetles, I assume.

We thank the reviewer for the suggestion and we made necessary modifications in the revised manuscript. However, also considering reviewer's previous comment on surface chemistry, we found it more intuitive to plot the traction force data normalized to PDMS glass replicas instead of glass. There are replicates in each leaf stage and the effect of replicates as random factors is low compared to that of beetles. Therefore, in order to plot the data relative to PDMS glass replicas, we have taken means of forces within the replicates and used them for statistical analysis (see results section and Fig. 5). We have also included the absolute force values wherever necessary and moved the original graph to the supplementary information.

Figure 4 and the connected results seem to be a specialised presentation of artefacts and as such rather should be placed in the SI than the main text. The logic of the manuscript greatly suffers from this.

We thank the reviewer for the suggestion. As suggested, we moved the figure 4 and the associated data in Table 1 to supplementary information (as Supplementary Fig. S5 and Table S2).

Figure 7 seems to be superfluous, as all details are known in literature and could just be referred to.

We agree with the reviewer that the details in the Figure 7 (Figure 6 in the revised manuscript) are already known in literature. However, we believe that the inclusion of this figure complements the discussion very well and helps in easing the understanding of the results for the general and broad readership of Royal Society Open Science. Therefore, we decide to include the Figure 7 (now Figure 6) as it is.

Minor corrections:

P7L56 replicates replicas

We believe that the term 'replicate' in this sentence is correct in a statistical sense as it refers to each sample in a single leaf stage. Nonetheless, this information is included in the revised manuscript to avoid confusion.

P9L10 the real high aspect ratios could be measured; rephrase

We thank the reviewer for the suggestion. The sentence is rephrased in the revised manuscript.

All in all, the presented results are interesting but it seems that the focus of the core findings is lost. This reviewer gets the impression that a lot of data is generated without providing a comprehensive story around it. In addition, the manuscript reads like incoherent, as if several parts have been written by different authors, so please have the manuscript reads again by an English speaker.

We thank the reviewer for finding our results interesting, and we are thankful to the reviewer for the detailed review and critical comments. Based on the reviewer's suggestions and after re-assessing the entire manuscript, we have made modifications in the sentence structure, added and removed content wherever necessary to maintain the focus of the story. We believe that the revised version of the manuscript reads well and provides a comprehensive story. We have also used a professional language editing service for the manuscript. We hope that the reviewer finds the revised manuscript satisfactory.

Reviewer: 2

Comments to the Author(s)

Dear Authors,

First of all, congratulation for such study.

The aims are well exposed, the statistics work is well conduct and despite the quantity of data, everything is understandable. I join the PDF of your manuscript only with few minor comments which are not affect the study itself.

We thank the reviewer for careful review of our manuscript and for the wishes. We are happy that the reviewer finds our manuscript interesting and well written.

I have only ONE remark that I would like to share to you. Something that I was a bit "frustrated" to not understand why those specific chose of plant and insect. Let me explain.

As you know, plants and insects are constantly interacting in complex ways through forest communities since hundreds of millions of years. For both mutualistic (such as pollination) and unilateral (such as herbivory) we are talking about a co-evolution, sometimes very specific between plant and insect species. Consequently, some plants and insects developed some characteristics that are specifics to their interaction.

In your introduction to your study, it looks like you choose one plant and one insect that are not specifically interact to each other and I would like to wonder why?

I full understand that some plant characteristics can be easier to analyse one some specific species, same for insect, maybe the one of this study is easier for you to handle. To me, it's not explicit in the introduction why to choose those species (both plant and insect) and it frustrated me to think that despite this experimentation, which is very interesting, this "interaction" of this insect walking on this leaf is not existing in real life.

This is the only point that I would like authors responds in their introduction in 1-2 sentences, please.

We thank the reviewer for the helpful suggestion and entirely understand his point. Therefore, we have explained in more detail the reasoning for the choice of the plant and animal species used in our

experiments. We have already explained our selection of species in the introduction and pointed out that this selection gives only a general insight into how the walking forces of insect vary with growth-induced ontogenetic changes in leaf cuticular structures. We have also mentioned that the Colorado potato beetles are readily available for experimentation, easy to experiment and that the attachment structures are similar to that of *Hevea brasiliensis*-specific pest, *Luprops tristis*. We have also extended this idea in the discussion section of the original manuscript. The potato beetle was also well studied for insect traction experiments on rough surfaces which makes it appropriate for comparison. We have added this information in the revised manuscript. In addition, we have included the information on why we chose the plant species in the introduction section of the revised manuscript. We hope that the reviewer finds this information sufficient.

From the pdf version of the manuscript:

P4-L39: *Only on that? There is only a dependance on the ability of insect to hold on the leaf? no other parameters really? Is there any reference to such statement please? :)*

We thank the reviewer for the observation. We agree with the reviewer and have changed the sentence accordingly in the revised manuscript.

P6-L26: *Can it affect the surface of the leaf for your experimentation or that has not real impact?*

We covered the cut ends of the leaves with petroleum jelly immediately after the leaves were cut and started the replication process in the minimum possible time. Also, as a standard protocol in our experiments, we cleaned the leaves with DI water and dried them with compressed air in a gentle manner. From our personal observations, we are sure that both the methods did not affect the microscale morphology of *Hevea brasiliensis* leaf surfaces.

P12-L41: *I believe this is already stated in the introduction in the aims of this study, maybe this sentence is not necessary here to be repeated*

We agree with the reviewer's comments and have removed the sentence in the revised manuscript.

P-12-L48: *This sentence is too long, you should split in 2 sentences. :)*

The reviewer was referring to the sentence – “This differed from the definition of leaf growth stages for the same species in earlier studies, which only involved macroscale morphological parameters or physiological data.”. We believe this sentence is fine as it is, which was also confirmed in the language check by a native speaker.

P15-L8-9: *I believe there is a mistake in the sentence. "Our finding The fact..." ???*

We thank the reviewer for the observation. We modified it accordingly in the revised manuscript.

Appendix C

Response to editor's and reviewers' comments

The comments are in *black italic font* and our responses are in normal blue font.

Associate Editor Comments to Author:

A few minor typographical changes to make, but otherwise you're all set - well done!

We thank the editor for his conditional acceptance of our manuscript for publication in Royal Society Open Science. We made the suggested changes wherever necessary.

Reviewer comments to Author:

Reviewer: 1

Comments to the Author(s)

This revision has much improved and I see no major obstacles in the way. Only a few small things: i) make sure that all Latin names are italics, also in the titles of the references. ii) Same for the parameters in the main text, R_c etc, which should also be italics.

Otherwise this is a nice contribution that I look forward to seeing in print!

We thank the reviewer for his/her detailed comments on the original version of the manuscript. The reviewer's comments helped us improve the manuscript. We have made sure that all the Latin names and the parameters in the whole text are in italics.

We are pleased that the reviewer is happy with our contribution.